# Timescales of motor memory formation in dual-adaptation

**Marion Forano**[ID]**, David W. Franklin**[ID]*

Neuromuscular Diagnostics, Department of Sport and Health Sciences, Technical University of Munich, Germany

* david.franklin@tum.de

**Data Availability Statement:** All data files are available from the DataDryad database (https://datadryad.org/stash/share/Fb1s3GZrvxi2pDc0MkjE0dLE3qH3oBWA3aPkV6wpfOk).

**Funding:** The author(s) received no specific funding for this work.

## Abstract

The timescales of adaptation to novel dynamics are well explained by a dual-rate model with slow and fast states. This model can predict interference, savings and spontaneous recovery, but cannot account for adaptation to multiple tasks, as each new task drives unlearning of the previously learned task. Nevertheless, in the presence of appropriate contextual cues, humans are able to adapt simultaneously to opposing dynamics. Consequently this model was expanded, suggesting that dual-adaptation occurs through a single fast process and multiple slow processes. However, such a model does not predict spontaneous recovery within dual-adaptation. Here we assess the existence of multiple fast processes by examining the presence of spontaneous recovery in two experimental variations of an adaptation-de-adaptation-error-clamp paradigm within dual-task adaptation in humans. In both experiments, evidence for spontaneous recovery towards the initially learned dynamics (A) was found in the error-clamp phase, invalidating the one-fast-two-slow dual-rate model. However, as adaptation is not only constrained to two timescales, we fit twelve multi-rate models to the experimental data. BIC model comparison again supported the existence of two fast processes, but extended the timescales to include a third rate: the ultraslow process. Even within our single day experiment, we found little evidence for decay of the learned memory over several hundred error-clamp trials. Overall, we show that dual-adaptation can be best explained by a two-fast-triple-rate model over the timescales of adaptation studied here. Longer term learning may require even slower timescales, explaining why we never forget how to ride a bicycle.

## Author summary

Retaining motor skills is crucial to perform basic daily life tasks. However we still have limited understanding of the computational structure of these motor memories, an understanding that is critical for designing rehabilitation. Here we demonstrate that learning any task involves adaptation of independent fast, slow and ultraslow processes to build a motor memory. The selection of the appropriate motor memory is gated through a contextual cue. Together this work extends our understanding of the architecture of motor memories, by merging disparate computational theories to propose a new model.

**Competing interests:** The authors have declared that no competing interests exist.

## Introduction

The generation of smooth and accurate movements requires predictive compensation of both internal and external dynamics, thought to arise through the formation of an internal model [1–4]. This model, a neural representation of our interaction with the environment termed here a motor memory, is formed through repeated practice and driven by error and reward signals: a form of motor adaptation [5–9]. Adaptation, or learning, of a movement or task allows us to continually build, recall and update if necessary, these motor memories. Having been learnt, a motor memory can remain for months or years [10,11]. Similar to visuomotor [12] and saccadic adaptation [13], adaptation to novel dynamics exhibits savings [13–15], the ability of prior learning to speed subsequent relearning, spontaneous recovery of an initially adapted state after de-adaptation [16,17] and interference, a conflict of adaptation between similar tasks [18–20]. Motor adaptation has been modelled by a two-state model with different timescales [16,21,22]. Within this framework, short-term motor memory is composed of one fast system that learns quickly but forgets quickly, and one slow system that learns slowly but retains more of the learning. Importantly, this architecture of motor memory is able to reproduce many aspects of motor adaptation, including interference and spontaneous recovery [16], but not savings [23].

However, this two-state model cannot account for dual-adaptation, as introducing a new task drives the unlearning of the first task [24]. Studying dual-adaptation, that is learning simultaneously two opposing tasks, aims to explore the type of adjustments that might occur during everyday life. In daily life, we consistently switch between different tasks, going from drinking a cup of tea to controlling our computer mouse. As we switch from one task to another, we need to adjust appropriately for the change in the dynamics of the external objects (cup) or sensorimotor transformations (to the computer screen). Like other single task models [6,25], the dual-rate model [16] lacks the ability to represent and switch between different motor memories. In the absence of specific contextual cues, our sensorimotor system is also unable to learn two opposing dynamics over the same workspace; a phenomenon termed interference [4,16,19]. However, effective contextual cues, such as physical or visual separation in state space [18,26,27], lead-in [28,29] or follow-through movements [30,31], enable participants to simultaneously adapt to opposing dynamics [4,18,27,32], indicating the formation of independent motor memories.

In order to integrate dual-adaptation, or the formation of independent motor memories, the dual-rate model was expanded to a single fast process and multiple slow processes gated by a binary cue representing the context [24]. This model proposes that these multiple slow processes act in parallel, forming a specific memory for each of the learned tasks. Within this model, interference between learning two opposing tasks is reduced, as the contextual cue gates a specific slow process, allowing its modification only, while avoiding alterations of other slow processes. However, this model [24] claims a single fast process which adapts to any errors regardless of the contextual cue. When switching between tasks, this single fast process de-adapts to one while adapting to another, thereby not contributing to the formation of distinct motor memories, which only arise through the slow process. Here we propose that a structure with a single fast process and multiple slow processes does not predict the existence of spontaneous recovery within a dual-adaptation paradigm. The combination of both fast and slow processes have been suggested to be necessary to account for spontaneous recovery [16,17,24,33,34]. Therefore, we directly test the proposed model by examining the presence or absence of spontaneous recovery during simultaneous adaptation to two opposing force fields.

## Results

In the present study, we assess the formation and the recall of motor memories while simultaneously learning two opposing force fields. Previous research [24] proposed that the one-fast-two-slow-state model of motor adaptation best fit the data of dual-adaptation. Here we simulate both the one-fast-two-slow-state model (our baseline model for comparison against all other models) and the two-fast-two-slow-state model of motor adaptation in order to determine whether these two models have specific differences in their predictions of the classic adaptation-de-adaptation-error-clamp experiment (A-B-error-clamp paradigm) [16]. Specifically, here we simulate these two models of motor adaptation for a dual-adaptation task in which each force field was associated with a cue, similar to our experimental design in which the contextual cue is a visual workspace shift [18]. As there are only two force fields and two contextual cues in dual-adaptation, we will consider two separate learning processes, whereas the neural system must have a much larger capability. These learning models update the motor state (representation of motor commands) on the next trial as a function of the current motor state and the error experienced on that trial [16]. In the simulation, the number of trials of the de-adaptation phase were adjusted such that the adaptation phase ended when the total motor state output returned to zero for the one-fast-two-slow-state model. The same number of trials was used for the other model.

The simulation of the two models predicts different motor outputs during the error-clamp phase (Fig 1). In the one-fast-two-slow model (Fig 1A), the single fast process (dashed magenta line) is continually updating to both opposing force fields, which results in an output close to zero (mean of both force fields). The total outputs (thick red and blue lines) are therefore almost entirely driven by the two slow processes (dotted lines). In this case, de-adaptation, where the total output is driven towards zero, requires the active unlearning of the two slow processes, erasing the motor memories of the previously learned tasks. In the final error-clamp phase, where the error is clamped at 0, the total output remains at zero as both the slow and fast processes have outputs close to zero. In contrast, the two-fast-two-slow model (Fig 1B) has separate slow and fast processes for each contextual cue and its respective force field. In this case, the total motor output for each cue (red or blue thick lines) exhibits a similar pattern to that seen for a single adaptation task [16]. In particular, adaptation to each force field occurs through the update of the respective slow and fast processes gated by the contextual cue. The total motor output in early adaptation is mostly due to the fast process (dashed line) whereas in late adaptation the slow process (dotted lines) contributes most. De-adaptation occurs primarily through updates of the fast process, such that the total motor output returns to zero. However at this stage the slow process still retains some of the learned motor memory. Finally, in the error-clamp phase both the slow and fast processes gradually decay to zero but at different rates. In contrast to the previous model, the slow process still retains part of the learned memory, resulting in a rebound of the total output towards the first learned task—a process called spontaneous recovery. Consequently, these two models make very specific testable predictions. If the one-fast-two-slow model is correct, we will see no spontaneous recovery in a dual-task paradigm. Conversely, if the two-fast-two-slow model is correct, we will find evidence of spontaneous recovery. In order to test the predictions of our two models, two experiments were conducted in which a total of twenty participants performed a dual-adaptation task.

### Experiment 1

**Experimental results.** Ten participants grasped a robotic handle (Fig 2A) and performed forward reaching movements in the same physical location while two contextual cues were

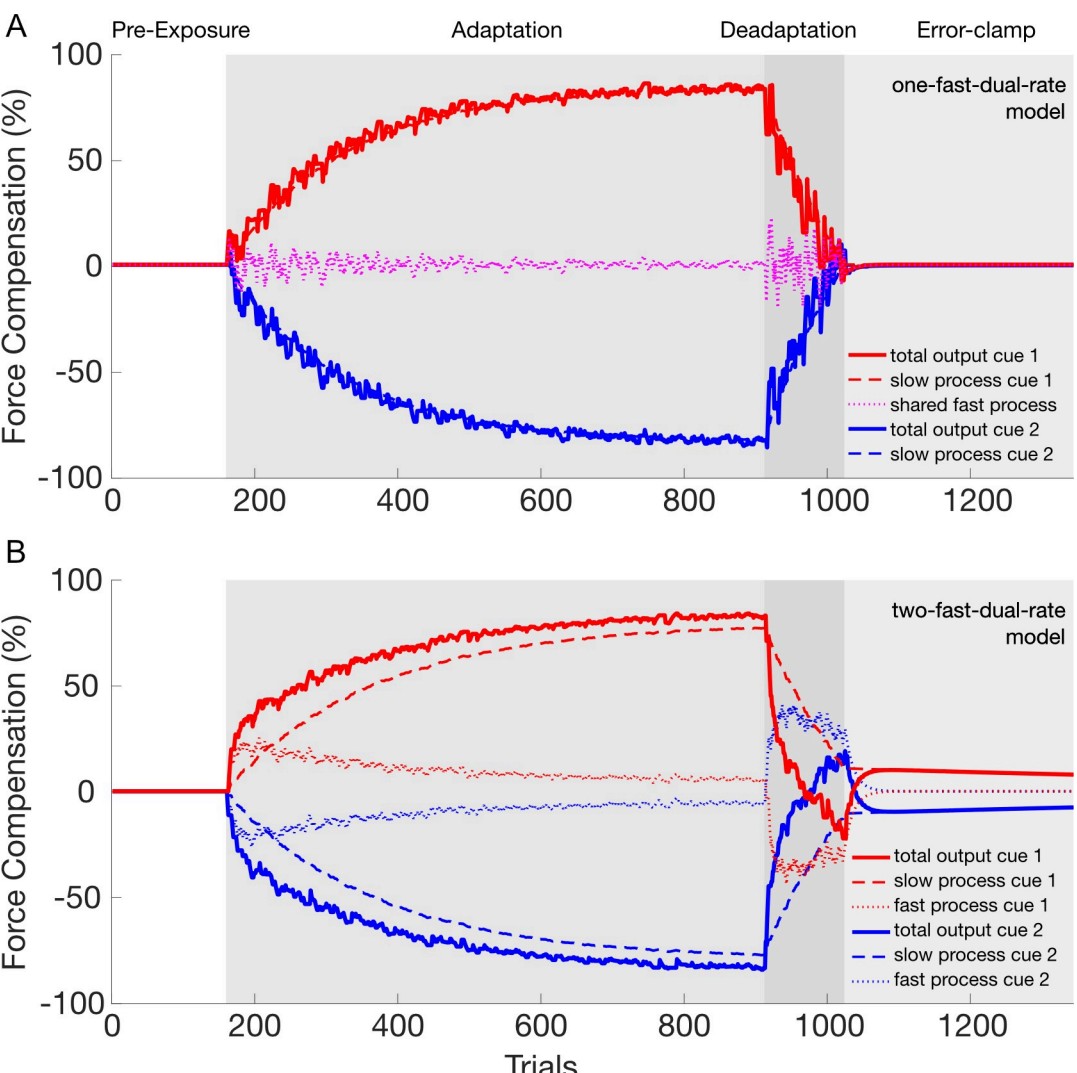

**Fig 1.** Simulation of the dual-rate model in dual-adaptation to compare a structure with one fast process [24] (**A**) against a structure with two fast processes (**B**). The model output of the contextual cue 1 (left workspace shift) and 2 (right workspace shift) are presented with red and blue lines, respectively. **A.** The dual-rate model of motor adaptation with one fast process and two slow processes. The total output for each contextual cue (red and blue thick lines) is composed of the summation of each slow process (red and blue dashed lines) and the single fast process (magenta dotted line). Note the absence of any spontaneous recovery. **B.** The dual-rate model of motor adaptation with two fast and two slow processes. The total output for each contextual cue (red and blue thick lines) is composed of the summation of each fast (red and blue dotted lines) and slow (red and blue dashed lines) process. Note that spontaneous recovery is revealed for both contextual cues in the error-clamp phase.

provided (visual feedback of the movement in the left or right half of the workspace, Fig 2B). The two contextual cues were linked to two opposing force fields in an A-B-error clamp design to examine the existence of spontaneous recovery (Fig 2C). We investigated differences in the kinematic error throughout the experiment using repeated measures ANOVA with main effects of stage (4 levels: early exposure, late exposure, early de-adaptation and late de-adaptation) and cue (2 levels). We found a significant main effect of stage ($F_{3,27} = 47.281$; $p < 0.001$) but no significant effect of cue ($F_{1,9} = 4.839$; $p = 0.055$. A similar repeated measures ANOVA was performed on the force compensation resulting in a significant main effect of stage ($F_{3,27} = 299.636$; $p < 0.001$) but no significant effect of cue ($F_{1,9} = 0.013$; $p = 0.912$). Further differences between specific stages in the experiment were examined using post-hoc comparisons.

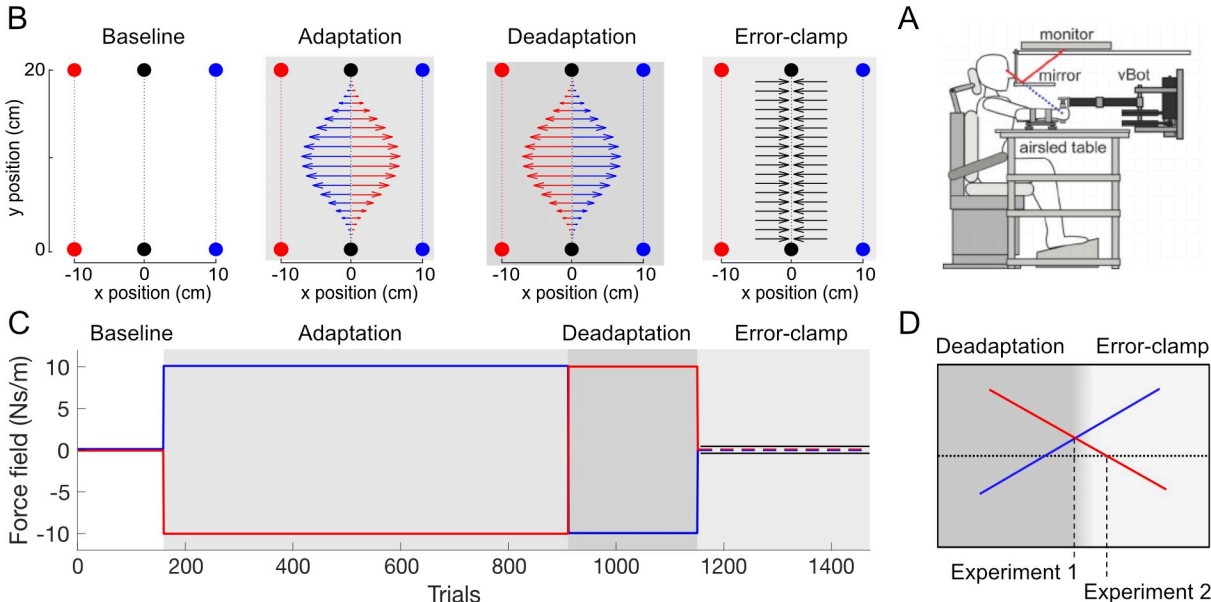

**Fig 2. Experimental Setup and Paradigm. A.** Workspace layout of the experiment. Participants always physically performed forward reaching movements in the center of the workspace (black). During these physical movements, the visual feedback (targets and cursor) was presented at one of two visual workspaces (red and blue) which acted as a contextual cue: -10cm offset (left workspace) set as cue 1 and +10cm offset (right workspace) set as cue 2. Red and blue colors are only used for illustration. On any one trial, only one of the visual workspaces was presented. In each phase of the experiment, different force fields were generated. In the pre-exposure phase, no external force was applied on the handle by the robot (null field trials). In the adaptation phase, two force fields were applied (CW and CCW), where each force field was always associated with one of the contextual cues (e.g. CCW force field for the left visual workspace and CW force field for the right visual workspace. In the de-adaptation phase, the association of the force fields to the visual cues was reversed (e.g. CW force field for the left visual workspace and CCW force field for the right visual workspace. In the error-clamp phase, all movements were performed in a mechanical channel for both contextual cues. **B.** Participants were seated and grasped the handle of the robotic manipulandum (vBOT) with their forearm supported by an airsled. Visual feedback of movements, displayed by the monitor, were viewed through a mirror so that they appear in the plane of movement. **C.** Temporal structure of the experiment. The different force field parameters depend on the experimental phase in order to create an A-B-error-clamp paradigm. In the error-clamp phase (light grey), the kinematic error is held at zero to assess spontaneous recovery. Trial numbers are shown for experiment 1. **D.** The participant-dependent transition from the de-adaptation to the error-clamp phase differed between the two experiments. In experiment 1, the participant's predictive adaptation (force compensation) for the two cues had to cross one another. In experiment 2, the participant's predictive adaptation was required to change sign (cross zero-line) for both cues.

In the pre-exposure phase, the movements were close to a straight trajectory with kinematic error (Fig 3A) and force compensation (Fig 3B) remaining close to zero. There was no difference in the force compensation between the two cues over the last 5 blocks (paired t-test: $t_9$ = 0.743; p = 0.476) and the force profiles for the two contextual cues were not distinguishable (Fig 3C).

In the adaptation phase, each contextual cue was associated with one of two curl force fields (clockwise and counterclockwise). As expected, initial trials in this adaptation phase exhibited large kinematic errors in the lateral directions, as the force field associated to each cue perturbed the movements in opposite directions (Fig 3A, red and blue curves). Throughout the adaptation phase participants gradually reduced their kinematic error from initial to final adaptation (post-hoc comparison: p<0.001) across both cues (Fig 3A). During this same phase, the force compensation (Fig 3B) increased gradually until it plateaued near 80% of perfect force compensation to each force field. A post hoc comparison showed an increase in force compensation between pre-exposure and final adaptation phases (p<0.001). The force profiles at the end of adaptation (last 10 blocks) demonstrate a bell-shaped profile similar to their forward velocity throughout the movement (Fig 3D). This pattern indicates a predictive compensation for both of the velocity-dependent curl force fields. This result, combined with

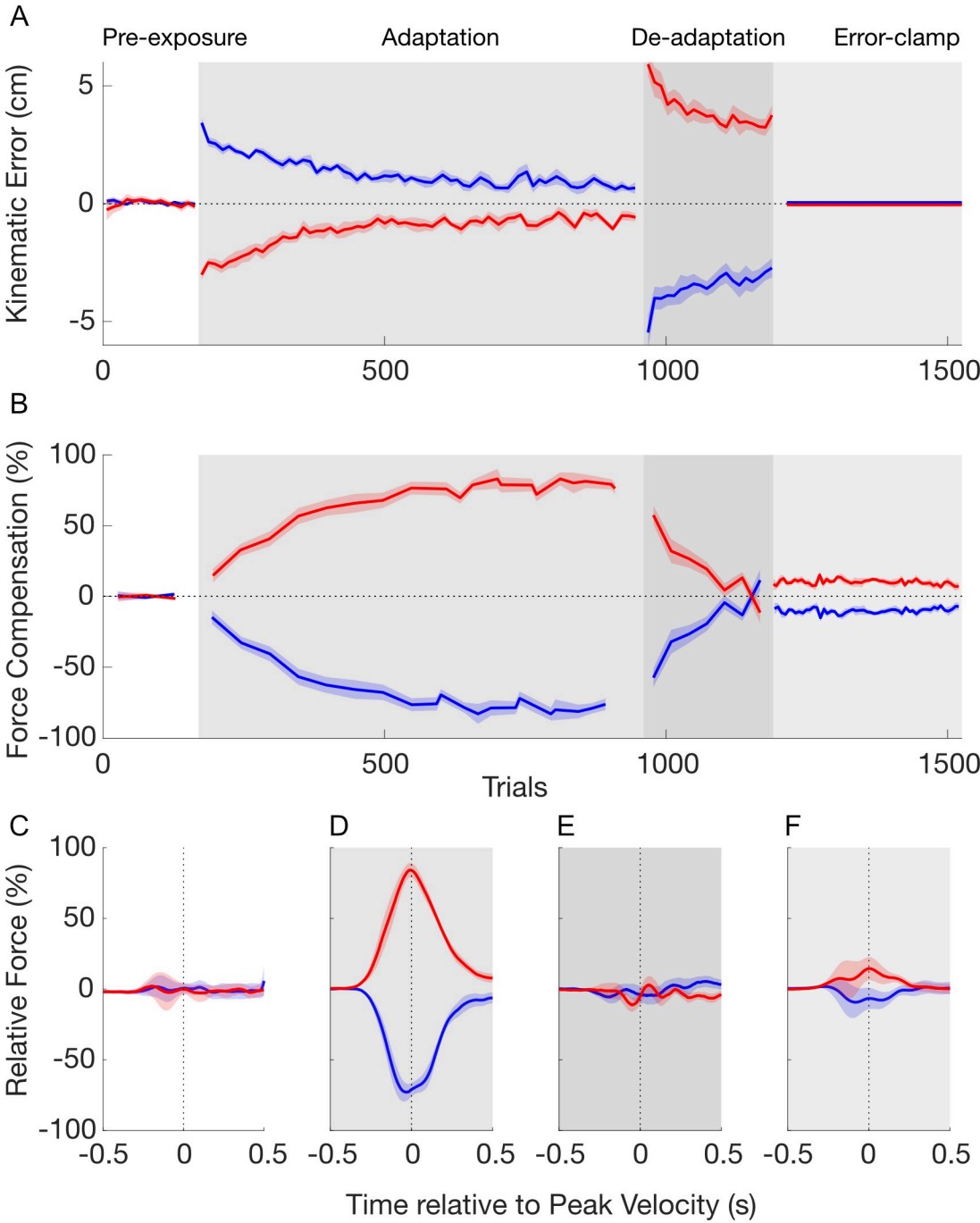

**Fig 3. Temporal pattern of adaptation to opposing force fields in experiment 1. A.** Mean of kinematic error over pre-exposure (white), adaptation (grey), de-adaptation (dark grey) and error-clamp (light grey) phases. The data of the contextual cue 1 (left visual workspace shift) and 2 (right visual workspace shift) are presented in red and blue lines, respectively. Shaded regions indicate the standard-error of the mean. **B.** Mean of force compensation. The force compensation for the two contextual cues is symmetrical due to the subtraction of the mean force compensation across the two cues. **C.** Force profiles on the channel wall as a function of movement time in the pre-exposure phase. The force values are shown as a percentage of perfect force compensation and aligned to peak velocity. For each cue, the mean of force compensation is taken from all trials. **D.** Force profiles for the last 10 trials (blocks) in the adaptation phase. **E.** Force profiles for the last 3 trials (blocks) in the de-adaptation phase **F.** Force profiles for all trials in the error-clamp phase.

the reduction in kinematic error, demonstrates that participants were able to adapt to the two opposing force fields simultaneously, supporting the finding that visual workspace location is a strong contextual cue for dual-adaptation [18].

In the following de-adaptation phase, the association between the force fields and contextual cues were flipped in order to reduce the total output of predictive compensation to zero. The initial trials in this phase show a large kinematic error in the opposite direction to that in the initial exposure trials (Fig 3A). Over the following blocks in the de-adaptation phase the kinematic error was reduced until it was similar in size (but opposite in direction) to the initial adaptation trials (post hoc comparison, p = 0.249). The force compensation (Fig 3B) rapidly decreased in this phase until there are no significant differences between the force compensation for the two contextual cues (paired t-test: $t_9$ = 0.409; p = 0.692). This is further supported by the force profiles of the end of the de-adaptation phase (Fig 3E), which show similar forces for each of the two contextual cues. Consequently, at the end of the de-adaptation phase there were no differences in the total predictive force output for the two contextual cues.

In the final error-clamp phase, channel trials clamp the lateral error to zero in order to assess the presence of any spontaneous recovery—rebound of the predictive force towards the adaptation exhibited in the first force field. Force compensation shows a quick rebound towards this first adaptation (Fig 3B) (post hoc comparison versus the last 3 blocks in de-adaptation phase: p = 0.016) followed by a negligible decay of the motor output throughout the twenty blocks. This evidence for spontaneous recovery is supported by the significant difference between the force compensation for the two contextual cues (paired t-test: $t_9$ = -6.538; p<0.001) across all twenty blocks. The force profiles during this error-clamp phase show predictive forces in the appropriate direction for compensation to the initial learned force field (Fig 3F). These force profiles peak near the time of peak velocity, but are reduced in size compared to those in the adaptation phase (Fig 3D). The force compensation results were analyzed after subtraction of the mean force compensation across the two cues for clarity (see Materials and Methods), but similar results were found when this subtraction was not performed (S1A Fig). Together, these results demonstrate that the participants maintain part of the memory of the initially learned task—arguing against the idea that dual-adaptation can be modelled by a one-fast two-slow-state model.

**Model fitting.**   Models from a family of learning-from-error equations were fitted individually to the participants' force compensation and compared using the Bayesian Information Criterion (BIC) on each participant's data (Fig 4). The individual BIC results are summarized in a frequency table according to their order of preference across participants (Fig 4A).

While Lee and Schweighofer [24] found that a one-fast-two-slow model best fit their experimental data, any evidence for spontaneous recovery argues against this model (Fig 1). Therefore, our previous experimental results (Fig 3) suggest that the two-fast-two-slow state model better explains dual-adaptation. We additionally compared BICs between these two models (S2A Fig) to check for evidence supporting the two-fast-two-slow state model. However, the BIC difference shows a better fit for the two-fast model for only three participants, whereas five participants have a better fit for the one-fast model, and two participants show no strong difference. With these opposing results, it is difficult to differentiate between the one-fast-two-slow and two-fast-two-slow models using only the BIC comparison.

To further investigate motor memory formation, and the presence of a single or multiple fast processes, we expanded our range of models to include both additional states (ultraslow and hyperslow) and a weighted gating of the contextual cues, producing a total of twelve models. Results show that the two-fast-triple-rate (binary-switch: 3 participants; weighted-switch: 3 participants), and the two-fast-weighted-switch-dual-rate (3 participants) models provided the best-fit equally to nine out of 10 participants (Fig 4A). Between these three selected models,

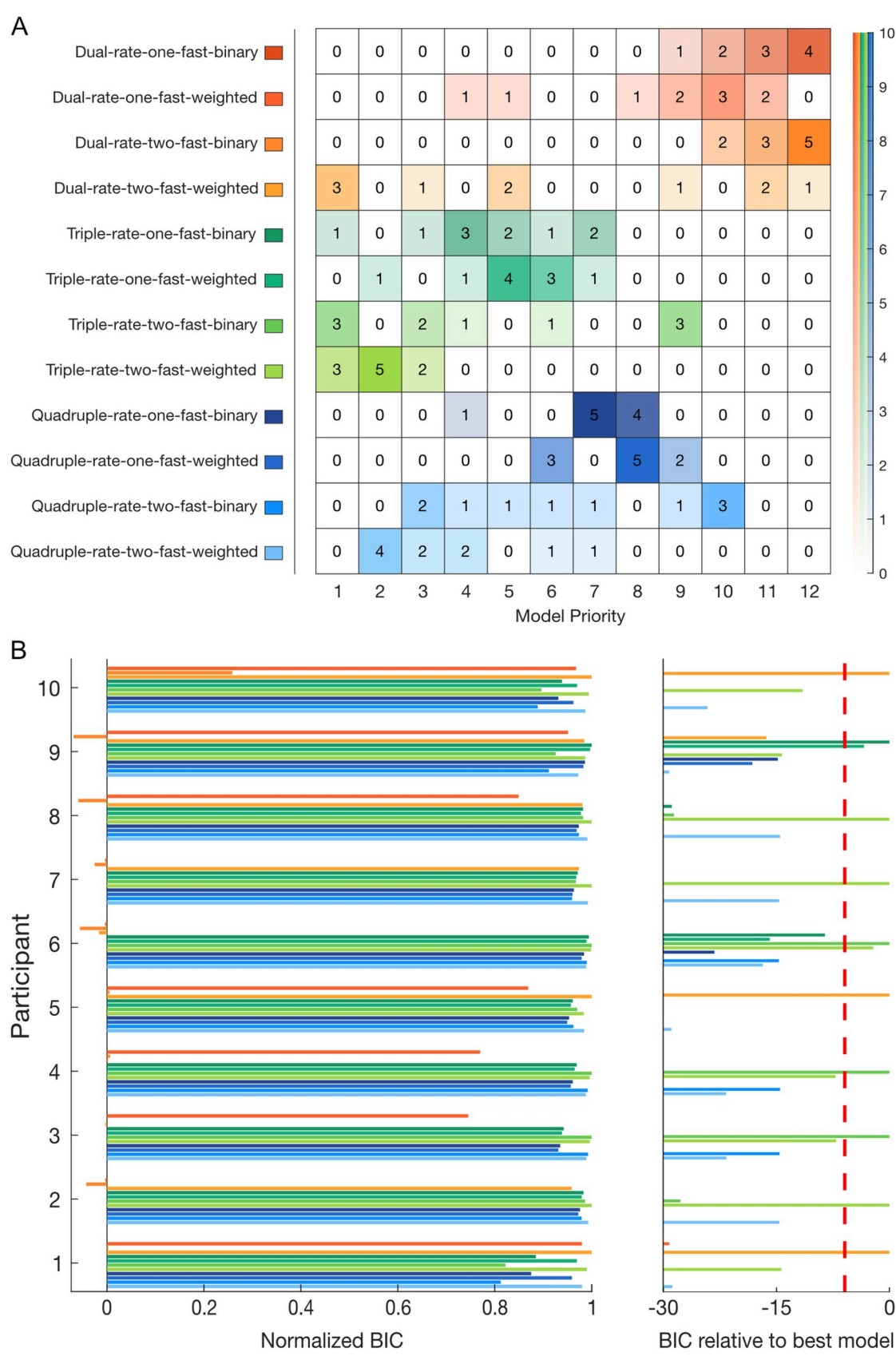

**Fig 4. Bayesian Information criterion (BIC) model comparison for individual fitting and frequency table for experiment 1. A.** Frequency table for each model (y-axis) by priority order (x-axis). This table represents the number of participants in which a given model was selected as the best-fit to the participant's data by BIC from first selected, to twelfth (last) selected. On the right-side, an opacity scale represents the number of participants. **B.** Individual BIC improvement for model comparison. The twelve models are differentiated by their improvement in BIC relative to the referent model (value of 0, left). For each participant, models are normalized to reveal the qualitative differences between each model and the best-fit model for that participant (value of 1). On the right side, the BIC improvement relative to the best-fit model is shown for the unnormalized values. Improvements in BIC from 2 to 6, 6 to 10 and greater than 10 are considered as a positive, strong and very strong evidence of a model fitting better than the other models, respectively. The red dashed line shows a BIC difference of 6 from the best-fitting model.

only the two-fast-binary-switch-triple-rate model was additionally selected second-best for 5 participants, including one participant where there is no strong evidence between this and the best-fit model. However, when we look at the overall preferences, it is clear that the family of triple-rate models performs better than that of the quadruple-rate models, which also performs better than the dual-rate models. Additionally, for both triple and quadruple-rate models, two fast processes are prioritized against a single fast process, suggesting again the existence of two fast processes. Here, it is still difficult to see a difference between the weighted-switch (either one or two-fast) models and their respective binary-switch models within each set of models (e.g. within the quadruple-rate models). Comparable patterns of priority were found for individual fits on the force compensation without the subtraction of the mean force compensation across the two cues (S3 Fig) except for the best-fit models where the two-fast-weighted-switch-quadruple-rate model was also selected.

As three participants were best-fit by the dual-rate model (dual-rate best-fit group), and seven participants were best-fit by the triple-rate model (triple-rate best-fit group), we examined these two groups of participants separately to see any adaptation differences, fitting both models to the mean participant results. The difference between the two groups is clearly seen in both the rate of adaptation and de-adaptation (Fig 5A and 5B). Additionally, a slight decay is observed during the error-clamp phase in the dual-rate best-fit group (Fig 5A), a phase that strongly contributes to the model fitting due to the extensive channel trials. The model simulations for both groups (Fig 5C–5F) illustrate the differences in the dual-rate and triple-rate models, however it is interesting that neither the dual-rate nor triple-rate models fit the slow adaptation seen for the dual-rate best-fit participants. However, when fitting both models (weighted switch) to the two groups of participants, strong evidence for the triple-rate model was found for the triple-rate best-fit group whereas no difference was found between the two models for the dual-rate best-fit group, suggesting that both models equally fit the experimental data (Fig 5G). Overall these results support the existence of a third timescale, and argue that this may be the case even for the three participants that were first best-fit by the dual-rate model.

## Experiment 2

**Experimental results.** Ten participants performed experiment 2. The major difference with experiment 1 was the criterion for ending the de-adaptation phase and proceeding to the error-clamp phase (Fig 2D), leading to a greater de-adaptation. Participants performed 240 ±26 trials in the de-adaptation phase compared to 211±30 trials in experiment 1. Overall the pattern of the results was similar to the first experiment. There were significant main effects of stage ($F_{3,27}$ = 42.292; p<0.001) but not cue ($F_{1,9}$ = 0.074; p = 0.792) for the kinematic error across the whole experiment. Similarly force compensation showed a main effect of stage ($F_{3,27}$ = 341.808; p<0.001) but not cue ($F_{1,9}$ = 0.007; p = 0.935). Initial movements in the pre-exposure phase exhibited little kinematic error (Fig 6A) and no difference between the two cues in force compensation (paired t-test: $t_9$ = 0.429; p = 0.678) over the 5 last blocks (Fig 6B) or force profiles (Fig 6C).

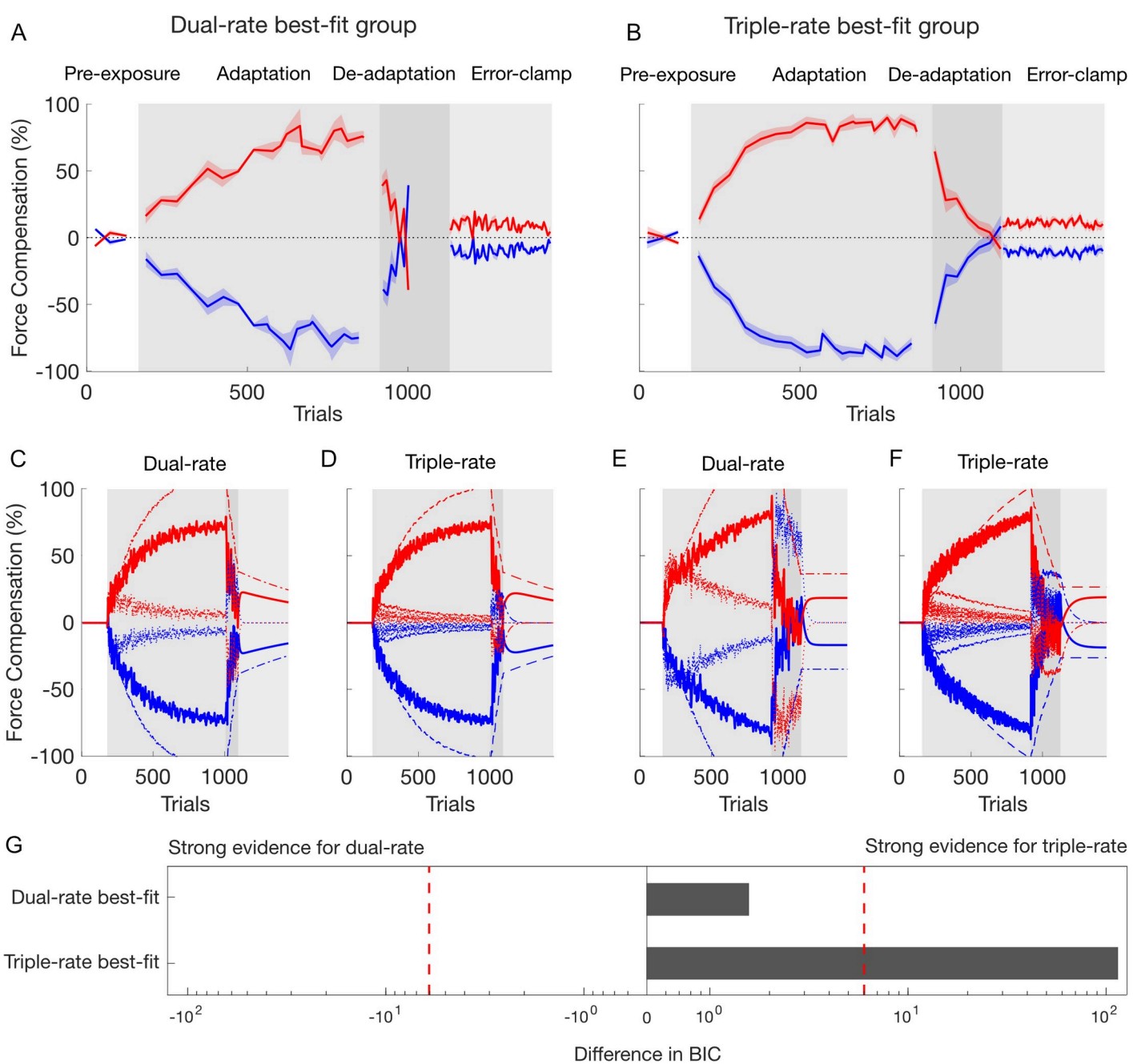

**Fig 5. Comparison between the 3 participants best-fit by the dual-rate model (left side) and the 7 participants best-fit by the triple-rate model (right side).** A. Mean of force compensation for the dual-rate best-fit group of participants. B. Force compensation for the triple-rate best-fit group of participants. C-F. Simulation of the dual and triple-rate models (weighted switch) from the fitted parameters of the dual-rate best-fit group (C and D respectively) and the triple-rate best-fit group (E and F respectively). The models were fit to the mean participant data rather than individual participants. Here, the de-adaptation phase contained 80 (C, D) and 208 (E, F) trials corresponding to the average trials number for the dual-rate best-fit and triple-rate best-fit groups respectively. The total output for each contextual cue thick red and blue lines) is composed of the summation of each process: fast (solid lines), slow (dotted lines) and ultraslow (dashed) process. G. BIC differences between dual-rate and triple-rate models for both groups. The red dashed line shows a BIC difference of 6 from the best-fit model.

Initial trials in the adaptation phase exhibited large lateral kinematic errors. These kinematic errors gradually reduced from initial to final adaptation (post hoc comparisons: p<0.001). During this same phase, the force compensation (Fig 6B) increased gradually,

 

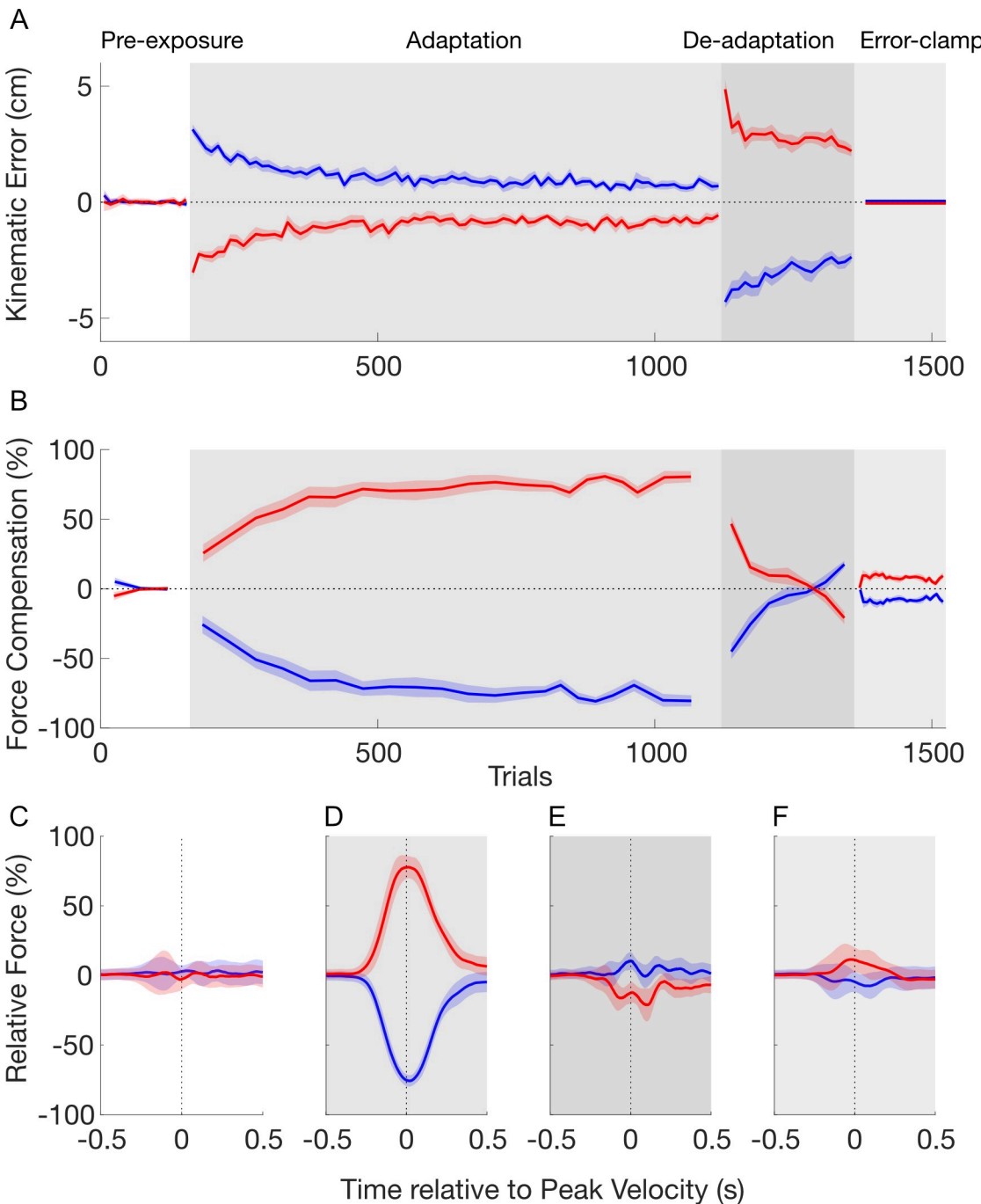

**Fig 6. Temporal pattern of adaptation to opposing force fields in experiment 2.** Plotted as experiment 1. **A.** Mean of kinematic error over pre-exposure (white), adaptation (grey), de-adaptation (dark grey) and error-clamp (light grey) phases. **B.** Mean of force compensation. **C.** Force profiles on the channel wall as a function of movement time in the pre-exposure phase. **D.** Force profiles in the adaptation phase. **E.** Force profiles in the de-adaptation phase **F.** Force profiles for all trials in the error-clamp phase.

plateauing around 80%. A post hoc comparison showed an increase in force compensation between pre-exposure and final adaptation phases (post hoc comparisons: p<0.001). The force profiles at the end of adaptation exhibit a bell-shaped profile (Fig 6D) indicating predictive compensation for both of the velocity-dependent curl force fields.

 

Initial trials in the de-adaptation phase showed a large kinematic error in the opposite direction to that in the initial exposure trials (Fig 6A). This error decreased until it was similar in size (but opposite direction) to the initial adaptation trials (post hoc comparisons: p = 1.000). In this experiment, de-adaptation required participants to deadapt independently to both cues resulting in a further de-adaptation compared to experiment 1. In fact, we observed force profiles which were already adapting to the opposing force field (Fig 6E), and found a significant difference in the force compensation between the two cues (paired t-test: $t_9$ = 4.479; p = 0.002).

In the error-clamp phase, the force compensation exhibited a quick rebound towards the initial adaptation (Fig 6B) (post hoc comparison versus the last 3 blocks in the de-adaptation phase: p<0.001) which was maintained over the rest of the phase. This evidence for spontaneous recovery is again supported by the significant difference between the force compensation for the two contextual cues (paired t-test for all blocks: $t_9$ = -4.400; p = 0.002). The force profiles show predictive forces in the appropriate direction for compensation to the initial learned force field (Fig 6F). Again, along with the results without the subtracted mean force compensation across the two cues (S1B Fig), the results of experiment 2 argue against the one-fast-two-slow-state model by demonstrating spontaneous recovery.

**Model fitting.** The twelve multi-rate models were individually fit to the data of experiment 2 and compared using BIC improvement on the individual participant's data. These results are summarized in a frequency table according to their order of preference (Fig 7A). We first compared the difference in BICs between the one-fast-two-slow [24] and two-fast-two-slow dual-rate binary models (S2B Fig). Here it is clear that the dual-rate model with two fast processes (binary) fits the data better than the dual-rate model with a single fast process (binary), with nine out of the ten participants displaying a significantly better fit.

When we examine across all twelve models, the two-fast-triple-rate model with weighted switching provided a better fit overall than all other models (5 participants compared to 4, for the similar model with binary switch and 1 for the two-fast-weighted-switch-dual-rate model. Participants 4 and 5 prioritized the two-fast-binary-switch-triple-rate model, but this shows no significant difference (0.7 and 0.8 in BIC respectively) with their respective weighted-switch model (Fig 7A), so both models are considered equally as first selected. Again, the overall pattern of priority showed strong evidence for the family of triple rate models compared to quadruple-rate. The family of dual rate models consistently placed last. Moreover there was a general preference of the two-fast processes models over their respective one-fast models. Similarly the weighted switch is generally prefered for the two-fast processes models compared to the binary switch. Similar but more variable results were found when fitted on the data with no subtraction of the mean force compensation across the two cues (S4 Fig), with a higher priority for two-fast-weighted-switch-quadruple-rate, as seen in experiment 1 (S3 Fig).

## Combined experimental results

**Individual experimental results.** The results from both experiments provide similar support for the existence of spontaneous recovery and the presence of multiple fast processes. Moreover, the distribution of force compensation across the phases of the experiment for individual participants shows a consistency between participants and across the two experiments (Fig 8). Therefore, the mean data accurately reflects the behavior of the individual data. All participants adapted to the first field-cue association (ranging between 60 to 100%). With the exception of one participant in the first experiment (participant 9), all participants were able to de-adapt or even to start learning the opposite field in the de-adaptation phase. Finally, in the error-clamp phase, 17 out of the 19 participants that de-adapt to the force field exhibit spontaneous recovery to the first learning.

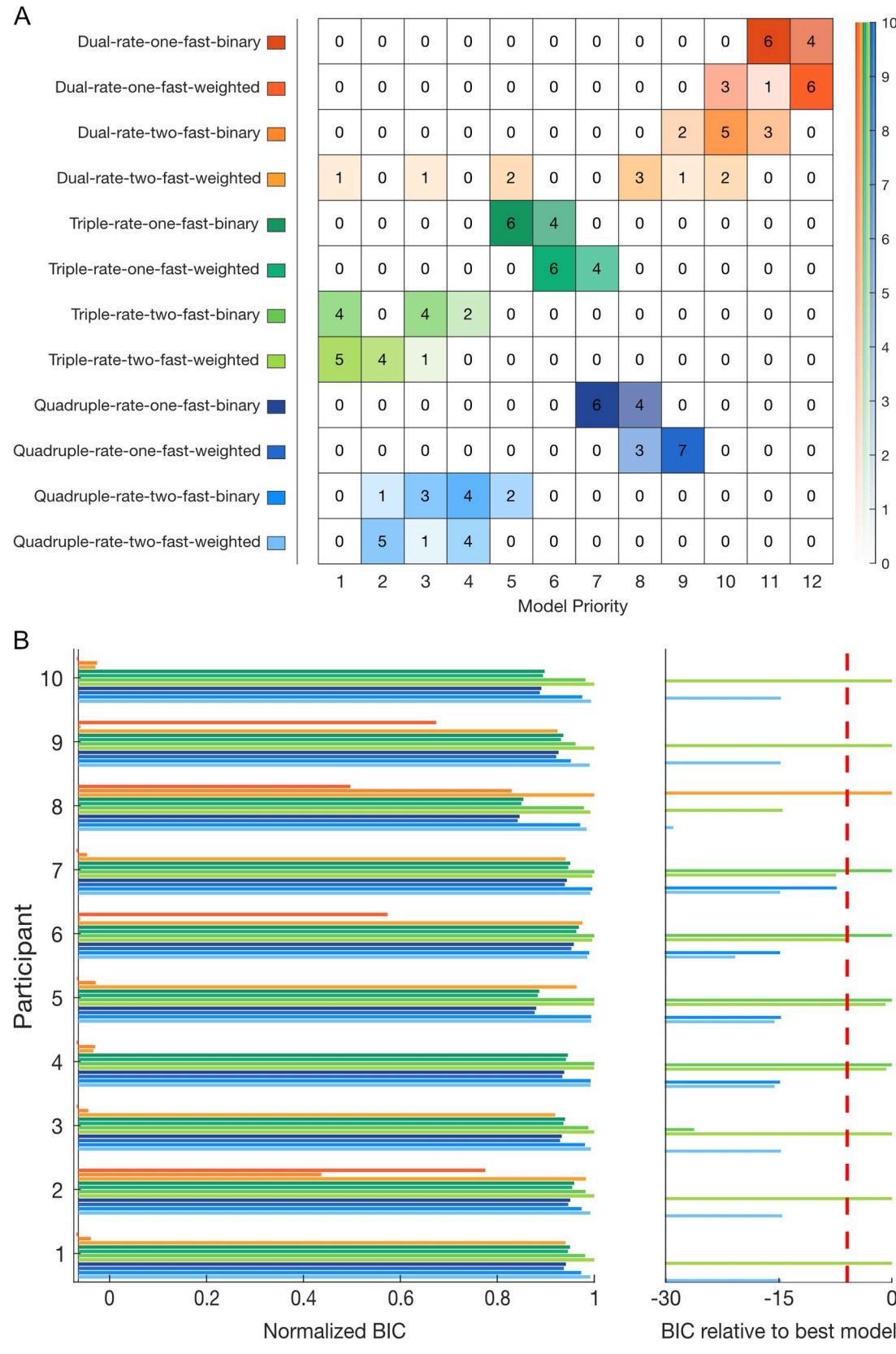

**Fig 7. Bayesian Information criterion (BIC) model comparison for individual fitting and frequency table for experiment 2.** Plotted as experiment 1. **A.** Frequency table for each model (y-axis) by priority order (x-axis). The right-side opacity scale represents the number of participants. **B.** Individual BIC improvement for model comparison.

**Model fitting.** To confirm which model or models best fit the two independent experiments, we combined the individual BIC from both experiments (Fig 9). Results showed a strong preference towards the triple-rate models and especially towards the two-fast-weighted-switch-triple-rate model, with a weighted contextual cue parameter preferably set between 0.8770 and 0.9538 (Table 1). Here, a preference of two fast processes is seen within any model type, arguing once more against the idea that dual-adaptation can be modelled by a one-fast two-slow-state model. Results for the fits performed without subtraction of the mean force compensation across the two cues (S5 Fig) are comparable to both individual experiments, with a higher variability between participants and indistinction between some models. Indeed, it is important to note that the order of preference in this fitting is less significant as some participants show tiny (participants 4, 5) of no significant differences (participants 1, 8, 13, 15, 18) between their first, second and sometimes third selected model (S3B Fig, S4B Fig). Together, the results demonstrate strong support for two fast processes and the existence of a third timescale—an "ultraslow" rate—that has a slower learning rate than the "slow" process. However, it is not clear whether weighted-switch models perform better than their corresponding binary-switch model. With the contextual cues studied in this experiment (e.g. visual location), we specifically chose a type of cue that is known to have a good contextual effect in dissociating opposite force fields [18] which would be expected to have a switching parameter closer to 1. As BIC punishes models with a higher number of parameters, the closer a weighted switching parameter is towards 1, the more likely BIC would select a binary switching model.

For comparison across the twelve models, the interquartile range and median of the best-fit set of parameters for each model were reported (Table 1). The median parameter values were

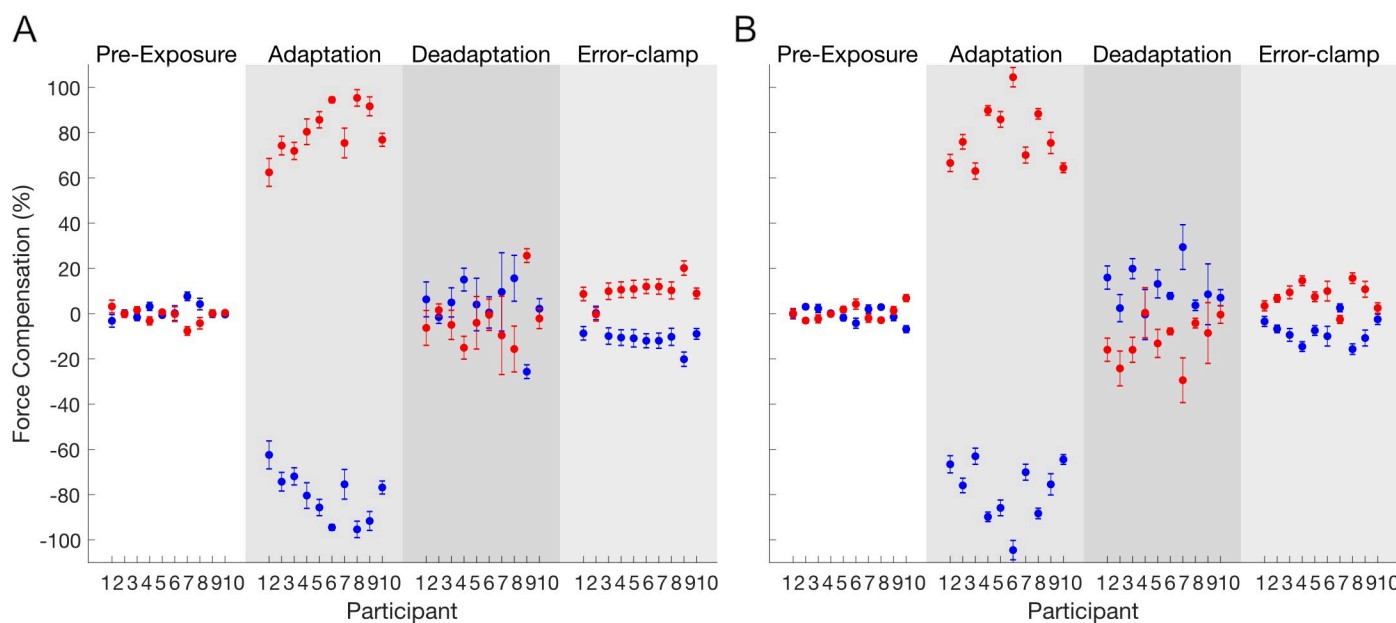

**Fig 8.** Distribution of individual participants in force compensation for experiment 1 (A) and experiment 2 (B). Each point represents the mean and standard error of force compensation over pre-exposure (white), adaptation (grey), de-adaptation (dark grey) and error-clamp (light grey) phases. For each subject, the force compensation for the two contextual cues is symmetrical due to the subtraction of the mean force compensation across cues.

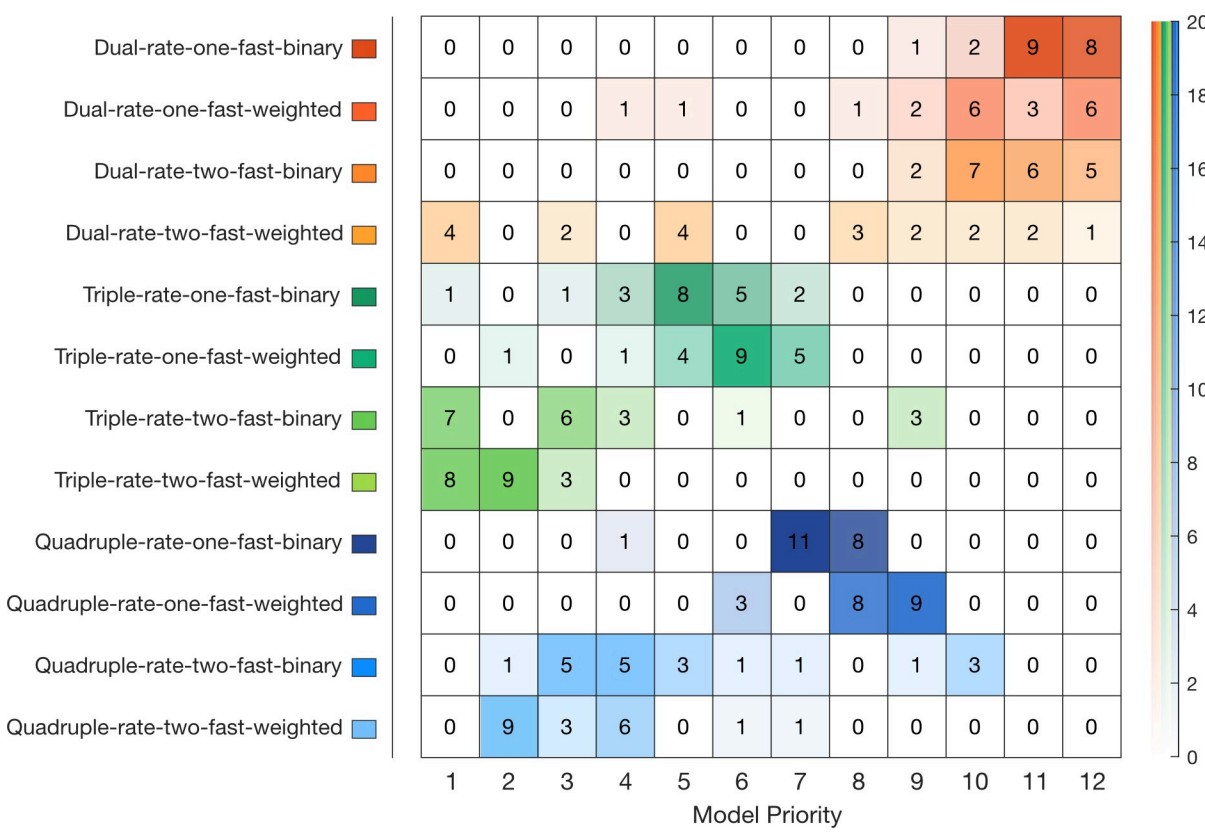

**Fig 9. Frequency table of the individuals Bayesian Information criterion (BIC) model according to their priority (order of preference according to BIC improvement) across both experiments 1 and 2.**

then used in model simulation where all twelve models were simulated with their best-fit parameters (S6 Fig). It can be seen that all triple and quadruple-rate models are able to capture the main features of the experimental data, including spontaneous recovery (Fig 10). In the case of these single-fast process models, spontaneous recovery occurs only due to the interplay between the slow and ultra-slow (or hyper-slow) processes. This again argues against the one-fast dual-rate model for explaining dual-adaptation.

In both experiments, although the best-fit model was the two-fast-weighted-switch-triple-rate model, there were individual differences in the BIC values, where for some participants the dual-rate model was selected as the best-fit model. Particularly, in experiment 1, although the overall results favored the triple-rate model, three of the ten individual BIC's preferred the two-fast-weighted-switch-dual-rate model (Fig 4). Our model recovery analysis (S7 Fig, S8 Fig), showed that the model fitting is able to accurately recover the binary or weighted switch and the number of fast processes, but often selects a model with fewer timescales. This suggests that it is likely that our model fitting will select a model with fewer timescales and may explain why the dual-rate model is occasionally selected for several participants. It is important to note that this also hints that the quadruple-rate model might best explain our experimental results, and that future experimental designs are needed to distinguish between these models.

**Decay.** Our experimental design allowed us to independently examine the retention rates (parameter $A$) by exploring the decay of the learned motor memory throughout specific decay blocks in the adaptation phase and in the error clamp phase at the end of each experiment (Fig 11). These are then compared to the decay that would be expected by a learning process with

**Table 1. Best-fit parameters of all models.**

| Model Type | | Retention rate | | | | Learning rate | | | | Switch |
|---|---|---|---|---|---|---|---|---|---|---|
| | | $A_f$ | $A_s$ | $A_{us}$ | $A_{hs}$ | $B_f$ | $B_s$ | $B_{us}$ | $B_{hs}$ | c |
| Dual-rate | 1 fast | **0.5000– 0.6067** (0.50000) | **0.9500– 0.9500** (0.95000) | | | **0.1000– 0.2204** (0.10000) | **0.0627– 0.1464** (0.08501) | | | |
| | | **0.5000– 0.6103** (0.50000) | **0.9500– 1.0000** (0.99818) | | | **0.1000– 0.1175** (0.10000) | **0.0200– 0.0822** (0.02509) | | | **0.7940– 1.0000** (0.86623) |
| | 2 fast | **0.7646– 0.8524** (0.81324) | **0.9500– 0.9799** (0.95000) | | | **0.1221– 0.3500** (0.26232) | **0.0200– 0.0370** (0.02000) | | | |
| | | **0.8061– 0.9201** (0.87438) | **0.9842– 1.0000** (0.99953) | | | **0.1084– 0.2939** (0.16488) | **0.0200– 0.0200** (0.02000) | | | **0.7488– 0.9062** (0.78895) |
| Triple-rate | 1 fast | **0.5000– 0.6024** (0.50000) | **0.9500– 0.9500** (0.95000) | **0.9995– 1.0000** (0.99989) | | **0.1000– 0.1046** (0.10000) | **0.0200– 0.0403** (0.02432) | **0.0048– 0.0085** (0.00564) | | |
| | | **0.5000– 0.6024** (0.50000) | **0.9500– 0.9500** (0.95000) | **0.9995– 1.0000** (0.99989) | | **0.1000– 0.1046** (0.10000) | **0.0201– 0.0403** (0.02432) | **0.0050– 0.0080** (0.00546) | | **0.9987– 1.0000** (1.00000) |
| | 2 fast | **0.5000– 0.6134** (0.50000) | **0.9500– 0.9500** (0.95000) | **0.9995– 1.0000** (0.99989) | | **0.1000– 0.1696** (0.10000) | **0.0200– 0.0200** (0.02000) | **0.0051– 0.0089** (0.00591) | | |
| | | **0.5000– 0.6854** (0.50000) | **0.9500– 0.9500** (0.95000) | **0.9995– 1.0000** (0.99990) | | **0.1004– 0.2244** (0.15124) | **0.0200– 0.0200** (0.02000) | **0.0059– 0.0092** (0.00759) | | **0.8770– 0.9538** (0.91995) |
| Quadruple-rate | 1 fast | **0.5000– 0.6024** (0.50000) | **0.9500– 0.9500** (0.95000) | **0.9500– 0.9979** (0.95899) | **0.9995– 1.0000** (0.99998) | **0.1000– 0.1046** (0.10000) | **0.0200– 0.0343** (0.02320) | **0.0000– 0.0019** (0.00000) | **0.0043– 0.0077** (0.00537) | |
| | | **0.5000– 0.6024** (0.50177) | **0.9500– 0.9500** (0.95000) | **0.9500– 0.9809** (0.95795) | **0.9995– 1.0000** (0.99997) | **0.1000– 0.1046** (0.10000) | **0.0200– 0.0296** (0.02305) | **0.0000– 0.0081** (0.00121) | **0.0023– 0.0059** (0.00506) | **0.9983– 1.0000** (1.00000) |
| | 2 fast | **0.5000– 0.6134** (0.50000) | **0.9500– 0.9500** (0.95000) | **0.9623– 0.9968** (0.97525) | **0.9995– 1.0000** (0.99997) | **0.1000– 0.1699** (0.10000) | **0.0200– 0.0200** (0.02000) | **0.0000– 0.0000** (0.00000) | **0.0044– 0.0070** (0.00558) | |
| | | **0.5000– 0.6838** (0.50051) | **0.9500– 0.9501** (0.95000) | **0.9562– 0.9932** (0.97564) | **0.9991– 1.0000** (0.99993) | **0.1000– 0.2203** (0.16488) | **0.0200– 0.0202** (0.02000) | **0.0000– 0.0000** (0.00000) | **0.0017– 0.0083** (0.00617) | **0.8770– 0.9538** (0.91993) |

The table shows the parameters of the cue switch (c), retention rate (A) and learning rate (B) for the fast, slow, ultraslow and hyperslow processes with both binary and weighted contextual cue switches. The interquartile range (25th and 75th percentiles, bold) and median were calculated across all subjects in both experiments. The grey shading highlights the best-fit model: the two-fast-weighted-switch-triple-rate model.

different retention rates ranging between 0.5 and 1.0. The first few trials of the error-clamp phase demonstrate the rebound in adaptation and do not allow us to assess decay. The following trials show a consistent force output throughout the rest of the error-clamp phase (Fig 11, orange lines). Comparison against the predictions of different retention rates suggest the presence of a process with a retention rate of at least 0.993, but with possibilities of higher values, especially for the results of experiment 1 (supports > 0.997). While these values could still represent the slow process as several studies have found values within this range [16,24,35–37], this also provides some support for the existence of an ultra-slow process in order to reflect actual motor learning. In addition, across the decay block channel trials, we find evidence of decay in both experiments (Fig 11, blue lines), supporting the existence of a retention factor between 0.9 and 0.95, although it is not possible to distinguish which process might be

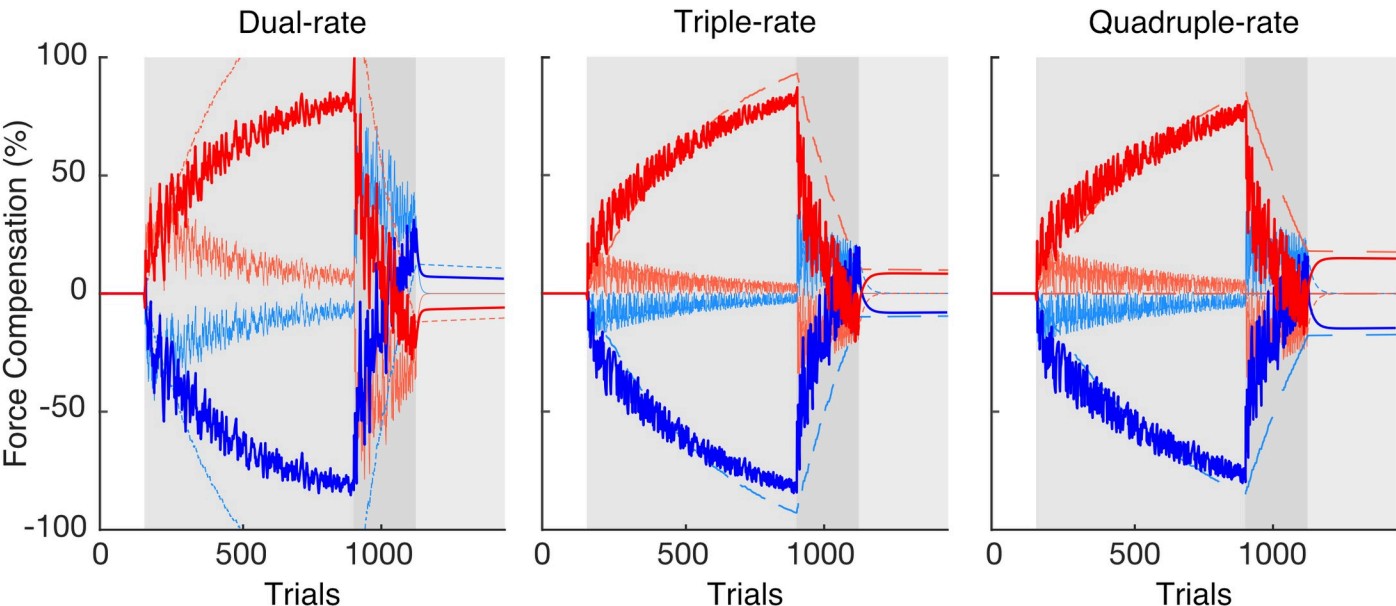

**Fig 10. Simulation of best-fit set of parameters (median across participants) for the dual, triple and quadruple-rate models with two fast processes and a weighted switch.** Here, the de-adaptation phase contained 220 trials (intermediate between experiment 1 and 2). The total output for each contextual cue (dark red and dark blue lines) is composed of the summation of each process (light red and light blue lines): hyperslow (long dashed), ultraslow (medium dashed), slow (dotted) and fast (solid) processes. Note that spontaneous recovery is revealed for both contextual cues in the error-clamp phase for triple and quadruple-rate models.

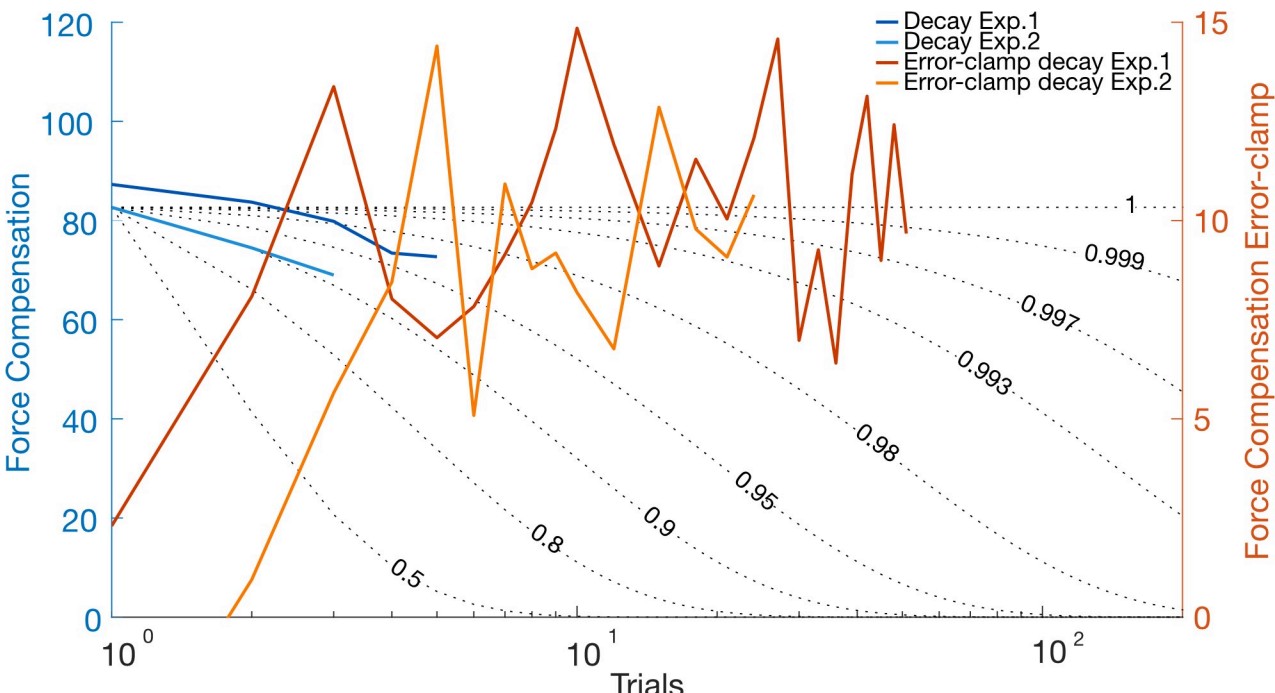

**Fig 11. Decay of force compensation (mean across participants) for the decay blocks in the adaptation phase (blue, left y-axis) and in the error-clamp phase (orange, right y-axis) for experiment 1 (dark colors) and experiment 2 (light colors).** Each data point in the error-clamp phase represents the mean of three trials. The scale for both y-axes was adjusted such that they can be compared to the simulated decay curves (black dotted lines). Nine retention curves were simulated with specific retention rates (0.5, 0.8, 0.9, 0.95, 0.98, 0.993, 0.997, 0.999 and 1), for comparison of retention rates and experimental data. Across all four experimental results, it is clear that a model must have a retention rate of greater than 0.993 for at least one of the processes in order to replicate these effects.

dominating this decay. These results differ from the best-fit model parameters but imply the existence of an "ultraslow" rate that results in the long retention of the learned model, as suggested in the study of Inoue and colleagues [37]. Although the retention rate of the ultraslow process is not much higher than that of the slow process in our best-fit parameters, the lower learning rate results in a maintained memory throughout the de-adaptation phase, so that the ultraslow process is expressed in the error-clamp phase (S6 Fig, two-fast-weighted-switch-triple-rate model). In these circumstances, the decay in the decay block is primarily governed by a mixture of the existing processes whereas the error-clamp decay is likely governed by the ultraslow process, as both fast and slow processes are expected to have been de-adapted during the de-adaptation and early parts of the error-clamp phase.

## Discussion

Here we investigated the structure of motor memory formation during force field adaptation. Specifically we tested the existence of one or multiple fast processes during adaptation by applying the A-B-Error-clamp paradigm [16] within dual-adaptation [18]. Participants initially adapted to two opposing force fields each associated with distinct contextual cues. After adaptation, the association between the contextual cues and the force fields were reversed, driving de-adaptation of the predictive force towards zero. Finally the kinematic error was clamped at zero to assess the presence of spontaneous recovery. In two experiments we found evidence for spontaneous recovery, supporting the two-fast-dual-rate model over the single-fast-dual-rate model. However, by comparing the fits of twelve models over a range of timescales, we found that a two-fast-triple-rate model best explained the experimental results. These results argue against previous models suggesting only a single fast process [24] and support the existence of longer timescales in motor adaptation.

The combination of both fast and slow processes have been suggested to be necessary to account for spontaneous recovery [16,17,24,33,34]. Our results, exhibiting spontaneous recovery of the initially learned force field for each contextual cue, support the existence of both two slow and two fast processes in motor adaptation. However, while simulations of the dual-rate model show that multiple fast processes are required to exhibit spontaneous recovery, spontaneous recovery could also occur through the combination of the slow process with an even slower one (e.g. ultraslow process). That is, the only requirement is multiple timescales for each contextual cue. In any case, during the error-clamp phase the remaining slower memory is revealed as the faster process decays to zero. In order to examine whether there are multiple fast processes, we fit the data with a variety of models using BIC for model comparison. Across both experiments, the best-fit model was a weighted-switch-triple-rate with two fast processes, providing further evidence that there are independent fast processes for each motor memory and arguing against previous findings [24].

The results of our experiments argue against a single generic fast process. However it is important to point out the differences with previous studies. While Lee and Schweighofer [24] used adaptation to opposing visuo-motor rotations, we examined dual-adaptation to opposing curl force fields. Thus the modality of adaptation is different, and could potentially explain the results. However we argue that a much larger difference is the experimental design in these two experiments. In order to learn the two opposing visuomotor rotations, Lee and Schweighofer [24] had the participants move to two different targets (different movement directions). Possibly, the participants do not even learn two separate motor memories but learn a single adaptation that generalizes between these two movement directions. Adaptation to different force directions for different movement directions has been shown many times during adaptation to a single force field [38–41]. If such adaptation processes also apply to visuomotor

rotation, then this would not be the ideal design for determining the presence of multiple fast processes as this design does not correspond to dual-adaptation.

Overall, models with weighted switching were preferred over their corresponding model with binary cue switching. To go further, we suggest that the switch parameter highly depends on the specific contextual cue used. That is, separation of the movements in visual space, as examined in our study, is a strong contextual cue for the learning of multiple dynamics [18,27] suggesting a weighting getting closer to one as seen in our results (between 0.8770 and 0.9538). Other cues, such as background color, object orientation [18], or static visual cues [28] which do not allow the formation of separate motor memories, might have weights closer to 0.5. Furthermore, dynamic cues, such as lead-in or follow-through movements [28,30,42,43], might be weighted in time and/or location. Our results, showing that interference occurs in dual-adaptation tasks even with a strongly effective contextual cue indicates that the contextual cue must be weighted rather than binary such as seen in our BIC. Indeed, several force field generalization studies [43–45] have shown that contextual cues do not serve as discrete switches but instead weight the contributions of associated motor memories. This matches the concept of multiple paired models for sensorimotor control [46]. In these models, each internal model or motor memory is selected by a responsibility predictor that determines its contribution to the overall motor output based on contextual information [2,47]. We suggest that our results argue for a similar framework in which the selection of each motor memory is based on a weighted input where the weight depends on the likelihood that this contextual cue predicts the next task.

Many previous studies have supported the dual-rate model to explain adaptation [16,17,24,33,34]. However, these studies have primarily focused only on short-term adaptation, examining at most several hundred movements. Consequently, in such short experiments an architecture consisting of two timescales may be sufficient to explain adaptation to a new task. Nevertheless, in daily life, we are often exposed to tasks or changes in environment over days or years. Indeed it commonly requires many months or years of practice to master a motor skill (e.g. using chopsticks or skiing). Once these skills are learned, people are able to recall them perfectly. In these cases, it is clear that a model of adaptation with only two timescales is insufficient to explain the formation of motor memories. Recent computational and behavioral studies have suggested the existence of longer timescales [37,48,49] such as an ultra-slow process in motor adaptation, which are further supported by our results.

In order to compare the time course of adaptation, the data was fit with dual, triple and quadruple models, where the model parameters were constrained. These constraints are a possible limitation, but necessary for the purpose of our study. Without constraints, model fitting finds solutions that do not test the proposed model structure. For example, a one-fast-rate model could assign fast-rate parameters to the slow processes in order to fit the data (thereby producing a two-fast rate model). Similarly, a dual-rate model can well fit data simulated under a triple-rate model, by only fitting fast and ultra-slow processes. However, the purpose of our study is not to simply ask how many timescales are necessary to fit the data. Instead, as we have extensive previous evidence [16,17,24,33,34] that there is at least a fast and slow process, our purpose is to see whether these two timescales can explain our current data, or whether additional timescales are also required. Therefore parameters were constrained to specific ranges (including all previous estimates of these values as outlined in Table 2) in order to test the specific model structures. A study with different questions, such as what is the smallest number of timescales needed to fit the data, may find that a dual rate model is sufficient under certain conditions.

Both our study and that of Inoue and colleagues [37] support the existence of longer timescales in adaptation and have shown little decay during the error-clamp phase after more extensive training phases. Our results, found with a design where participants initially adapted

**Table 2. Overview of relevant literature for parameter estimation.**

| Study type | Literature | Parameters | | | | | |
|---|---|---|---|---|---|---|---|
| | | $A_f$ | $A_s$ | $A_{us}$ | $B_f$ | $B_s$ | $B_{us}$ |
| Single Adaptation | Smith et al., 2006 (with 95% confidence intervals) | **0,59** (0,43–0,76) | **0,992** (0,990–0,994) | | **0,21** (0,10–0,35) | **0.02** (0.013–0.025) | |
| | Inoue et al., 2010 - Prism adaptation | **0.834** | **0.994** | | **0.182** | **0.0379** | |
| | | *0,7 <* **0,914** *< 0,95* | **0.995** | 0.9996 | *0 <* **0.202** *< 0,3* | **0.0612** *< 0,10* | **0.033** *< 0,033* |
| | Vaswani & Shadmehr, 2013 - Experimental data, Fit from of Inoue et al., 2010 | **0,876** | **0,995** | | **0,315** | **0,056** | |
| | | **0.69** | **0.977** | 0.991 | **0.189** | **0.147** | **0.013** |
| | Joiner & Smith, 2008 | **0.85** | **0.998** | | **0.11** | **0.021** | |
| Dual—adaptation | Lee & Schweighofer, 2009 - Visuomotor rotation (with 95% confidence intervals) | **0.8251** (0.6338–0.9767) | **0.9901** (0.9876–0.9986) | | **0.3096** (0.1585–0.5118) | **0.2147** (0.0582–0.2729) | |

The table shows the set of parameters of the retention rate (*A*) and learning rate (*B*) in multiple timescale models and their used constraints (italics) from the best-fit model to the experimental data. For Inoue (2010), both the dual-rate and triple-rate (best-fit) model parameters are reported. In Vaswani & Shadmehr (2013), both their best-fit model (top row) and those from re-fitting their model to Inoue (2010) are also shown (bottom row). The best-fit parameters are shown in bold and, where present, their 95% confidence intervals are indicated within the brackets.

to a force field for over 750 trials, suggests a triple-rate model of adaptation. However, we expect that for studies with longer training periods, the number of active timescales would increase. That is, we propose that there is a continuum of timescales in human motor learning, where the active number of timescales used for any given task depends on the overall time over which such a task is practiced (minutes, hours, days, or years) and the relative relevance of such learning for the future [50–53]. In this work we consider only one aspect of motor learning, that of learning the temporal pattern of muscle activations that produces a specific pattern of endpoint forces as studied in force field adaptation. However, many studies have examined how other factors of motor skill adaptation, for example learning the feedback control policy varies with experience [25,54–58]. While these studies have yet to examine adaptation of these other factors in terms of multiple learning rates, we suggest that these multiple timescales of adaptation will also be exhibited within other components of skill learning, with longer timescales underpinning long term learning. These longer timescales, with slow learning rates and almost no decay, act to protect the motor memories and could explain why we never forget some tasks, such as riding a bicycle.

One major question is where such learning timescales might be implemented within the sensorimotor system. While many studies have suggested that the cerebellum is necessary for initial adaptation [59,60], recent papers have proposed the primary motor cortex (M1) as the storage of these motor memories [59,61]. That is, there could be a shift from the cerebellum to M1, or that the fast process occurs within the cerebellum while the slow process occurs within M1. Further support has come from the finding that cerebellar patients adapt better to slow rather than abrupt changes in dynamics, suggesting the involvement of the cerebellum in the fast adaptation processes [9,62]. Moreover, support for longer timescales in motor adaptation has also been brought from neuroimaging studies. Specifically, different brain networks have been shown to be activated in at least three different timescales in dual-adaptation [63] with distinct regions of the cerebellum involved in both the fast and slow timescales. Accordingly, perhaps learning on multiple timescales takes place across multiple regions rather than shifting from one region to another.

The cerebellum has been implicated in storing multiple internal models [63–65]. Indeed, fMRI decoding has shown separate representations for different visuomotor mappings, with distinct substrates for each rotation in both the supplementary motor area and cerebellum [66]. However, none of these studies have been able to establish the existence of distinct representations for the fast timescale. Our results predict distinct representations due to the multiple fast processes, but the degree to which these might be spatially separable is unknown. Recent work has shown that rapid force field adaptation does not require changes in connectivity [67] or neural tuning [68], but instead may result from changes in the output-null subspace between the premotor (PMd) and M1 areas in order to produce rapid adaptation to the changed dynamics [67]. In this case, distinct spatial representations of multiple fast processes may not be detectable with current neuroimaging techniques.

To conclude, we expand the concept of the dual-rate model [16] into the framework of multiple paired models for sensorimotor control (MOSAIC model; [46]). Originally, the architecture of the MOSAIC model allows the learning of multiple paired inverse and forward models, by selecting the appropriate module (or modules) for the current environment. Built from Bayesian theories, this model uses a responsibility predictor to switch between (or combine) relevant modules based on contextual information [2,47]. We extend this framework by suggesting that these paired models are formed through adaptation on multiple timescales (e.g. fast, slow and ultraslow processes) (Fig 12). Whether or not the inverse and forward models are represented on different timescales [69] is not yet established. However, our results demonstrate both the existence of independent fast processes for each motor module and the presence of longer timescales in motor adaptation.

## Materials and methods

### Participants

Twenty-three force field naive participants with no known neurological disorders participated in two experiments. Three participants (one of experiment 1 and two of experiment 2) were removed for further analysis as they were not able to simultaneously adapt to the two force fields. In total 10 participants data were analyzed for both experiment 1 (n = 10, 4 female, 6 male, age 28±0.9 years) and experiment 2 (n = 10, 5 female, 5 male, age 26±1.8 years). Each participant only participated in one of the two experiments. All participants were right-handed based on the Edinburgh handedness questionnaire [70], and provided written informed

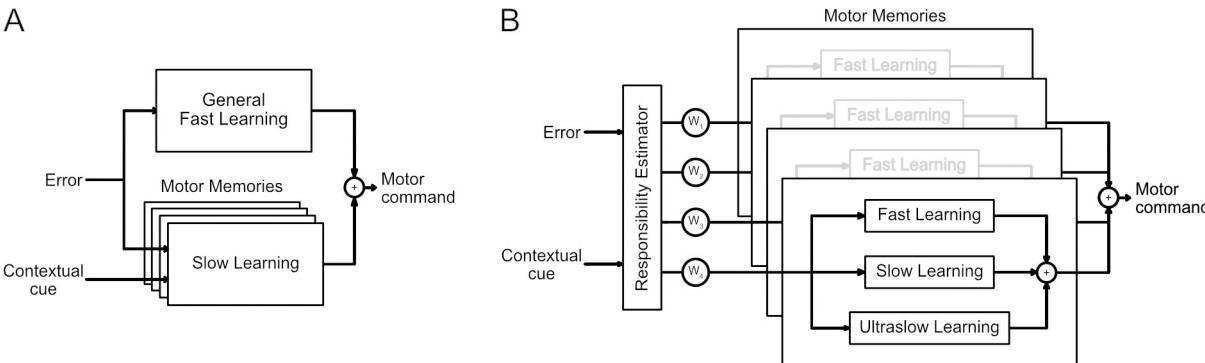

**Fig 12. Models of motor learning. A.** The dual-rate model with multiple slow processes acting in parallel and a single general fast process as proposed previously [24]. **B.** Our proposed model of adaptation consists of multiple parallel motor memories, each involving a fast, slow and ultraslow process, all weighted by a responsibility estimator.

consent before participating. The Ethics Committee of the Medical Faculty of the Technical University of Munich approved the study.

## Apparatus

Participants grasped the endpoint handle of a vBOT planar robotic manipulandum and virtual reality system [71], with their forearms supported against gravity using an air sled. The participants were seated in a custom adjustable chair in front of the apparatus, strapped with a four-point seat belt to reduce body movement. The vBOT system is a custom built robotic interface that can apply state-dependent forces on the handle while recording the position and velocity in the planar workspace (Fig 2A). This horizontal plane was located approximately 15 cm below the shoulder. A six-axis force transducer (ATI Nano 25; ATI Industrial Automation) measured the end-point forces applied by the participant on the handle. Joint position sensors (58SA; Industrial encoders design) on the motor axes were used to calculate the position of the vBOT handle. Position and force data were sampled at 1kHz. Visual feedback to the participants was provided horizontally from a computer monitor fixed above the plane of movement and reflected via a mirror system that prevented visual feedback of the participants' arm. The visual feedback, provided in the same plane as the movement, consisted of circles indicating the start, target and cursor positions on a black background.

## Protocol

Participants initiated a trial by moving the cursor representing the hand position (red circle of 1.0 cm diameter) into the start position (grey circle of 1.5 cm diameter) located approximately 20 cm directly in front of the participant. This start position turned from grey to white once the cursor entered it. Once the hand was within the start position for a random time between 1 and 2s, a go-cue (short beep) was provided signaling participants to initiate a reaching movement to the target (yellow circle of 1.5 cm diameter). The target was located 20.0 cm directly in front of the start position. The movement was considered complete when the participants maintained the cursor within the target for 600ms. After each trial, the participant's hand was passively driven to the start position while the visual feedback regarding the success of the previous trial was provided. Successful trials were defined as trials that did not overshoot the target and with a peak speed between 52 and 68 cm/s. On these trials, the participants received positive feedback (e.g., "great" for peak speeds between 56 and 64 cm/s or "good" for peak speeds between 52 and 56 or 64 and 68 cm/s), and a counter displayed on the screen increased by one point. In contrast, messages of "too fast" or "too slow" were provided when the peak speed exceeded 68 cm/s or did not reach 52 cm/s, respectively. Finally, "overshoot" feedback was provided when the cursor overshot the target (y-axis) by more than 1 cm. Movements were self-paced, as participants were able to take a break before starting the next trial. Short breaks were enforced after approximately 130 trials throughout each session.

Participants were instructed to reach naturally to the target after the go-cue on each trial while the dynamics of the environment were generated by the vBOT. During each movement, the vBOT was either passive (null field), produced a clockwise (CW) or counterclockwise (CCW) velocity-dependent curl force field, or produced a mechanical channel (error-clamp). For the velocity-dependent curl field [72], the force at the handle was given by:

$$\begin{bmatrix} F_x \\ F_y \end{bmatrix} = \begin{bmatrix} 0 & -k \\ k & 0 \end{bmatrix} \begin{bmatrix} \dot{x} \\ \dot{y} \end{bmatrix} \tag{1}$$

where $k$ was set to either $\pm 10$ N·m$^{-1}$·s, with the sign of $k$ determining the force field direction

(CW or CCW). On a channel trial, the reaching movement was confined to a simulated mechanical channel with a stiffness of 6000 N·m$^{-1}$ and damping of 2 N·m$^{-1}$·s acting lateral to the line from the start to the target [6,73]. As these trials clamp the lateral error at zero, they are also referred to as error-clamp trials [16].

## Experimental paradigm

Participants were required to adapt to an A-B-Error-clamp paradigm [16] within dual-adaptation. The A-B-Error-clamp paradigm is a sequential presentation of an adaptation phase (adapting to field A), a de-adaptation phase (presentation of opposing field B), and an error-clamp phase (assessment of any spontaneous recovery). Here the A-B-Error-clamp paradigm was done within dual-adaptation, where participants simultaneously adapt to two opposing force fields. To allow adaptation to two opposing force fields (CW and CCW), two appropriate contextual cues were used. These cues were the workspace visual location [16,18] either to the left (-10cm from the sagittal axis) or to the right (+10cm from the sagittal axis) of the screen. The physical hand location (proprioceptive location) was the same for both cues, without any shift from the sagittal axis (Fig 2B). Experiments began with a pre-exposure phase where movements with both contextual cues occurred in the null field. Within the adaptation phase one contextual cue (e.g. +10cm visual shift) was associated with one force field (e.g. CW), whereas the other contextual cue (e.g. -10 cm visual shift) was associated with the other force field (e.g. CCW) (Fig 2B and 2C). In the de-adaptation phase the opposing force fields were then applied to movements for each contextual cue (e.g. CCW for the +10 cm visual shift & CW for the -10cm visual shift) (Fig 2B and 2C). Finally, error-clamp trials were applied for both contextual cue movements. The experiments were counterbalanced such that in each experiment, half of the participants experienced the adaptation phase with contextual cues matched to one set of force field directions, whereas the other half of the participants had contextual cues matched to the opposing force field directions.

Experiments were built using blocks of trials where each block was composed of 16 trials: 8 with one contextual cue and 8 with the other contextual cue. For most of the experiment, for each contextual cue, seven of the movements were performed in either a null, CW or CCW force field (depending on the experimental phase) while one was performed in the error-clamp. Within a block trials were pseudo randomized.

**Experiment 1 (n = 10).** Ten participants started with 10 blocks in the pre-exposure phase (160 trials). This was followed by 47 blocks in the adaptation phase (752 trials). Within this adaptation phase, after the first 25 blocks, we modified the block structure in order to assess the memory decay for each contextual cue (blocks 26–45). Within these twenty blocks, 5 blocks assessed the decay for one contextual cue (+10 cm visual shift) whereas 5 blocks assessed the decay for the other cue (-10 cm visual shift). These decay blocks alternated between the two contextual cues with a normal block of trials interspersed between them. A decay block contained an exposure trial of a specific contextual cue followed by a row of 5 error-clamp trials with the same contextual cue. This row of trials was then always followed by an exposure trial of the opposite contextual cue. After these twenty memory decay blocks, two blocks were performed with the original block structure (blocks 46–47). Next, a de-adaptation phase was performed in which the opposite force field was now associated with each contextual cue. The length of the de-adaptation phase was dependent on each participant's performance and could vary between 4 and 25 blocks (64–400 trials). The exact point at which this phase was terminated depended on the difference between the mean of the force compensation measure (see analysis) of the last three error-clamp trials for each contextual cue. These force compensation values were calculated online during the experiment. Once this difference

switched sign (became negative) the de-adaptation phase was ended at the end of the current block of trials (Fig 2D). Finally participants experienced the error-clamp phase for 20 blocks (320 trials) in which all trials were error-clamps. Within these final blocks, eight error-clamp trials were performed with one contextual cue (+10 cm visual shift) whereas the other eight trials were performed with the other contextual cue (-10 cm visual shift) in a pseudo-randomized order.

**Experiment 2 (n = 10).**    The second experiment followed a similar design to the first one, but with specific differences outlined here. After the pre-exposure phase (10 blocks), the adaptation phase lasted for 60 blocks (960 trials). Within the adaptation phase, the memory decay blocks were again applied (blocks 41–50), but with no interspersing normal blocks. Within these memory decays blocks, three error-clamp trials were performed in a row directly after the normal exposure trial for the specific contextual cue. The de-adaptation phase could vary between 4 and 20 blocks (64–320 trials) depending on each participant's performance. In this experiment, the phase was ended once the mean of the last three force compensation values on the clamp-trials for each cue crossed zero (Fig 2D). Finally the error-clamp phase lasted for 10 blocks (160 trials). This second experiment was added to examine the behavior in the error-clamp after a stronger and cue-independent de-adaptation. The number of error clamp trials was also reduced as little effect of the trial number in this phase was found in the first experiment.

## Analysis

Data were analyzed offline using MATLAB (2018b, The MathWorks, Natick, MA). Individual trials were aligned on peak velocity, and clipped between -500 ms and +500 ms. All measurements were low-pass filtered at 40 Hz using a $10^{th}$ order zero-phase-lag Butterworth filter (filtfilt). Over the course of the experiment, the lateral force measured can vary due to the natural drift of the mass of the arm over the airsled. In order to improve our sensitivity of the measurement of the lateral force, we subtracted the mean offset in lateral force for each participant measured before the onset of movement from the total force trace. More precisely, the mean force was calculated over all trials for each participant between -250 and -150 ms prior to the movement start. In order to quantify adaptation throughout the experiments, three measures were calculated: kinematic error, force compensation and force profiles.

**Kinematic error.**    For each non-channel trial (null or curl field trials), the maximum perpendicular error was calculated and used as a measure of kinematic error. The maximum perpendicular error is the signed maximum perpendicular distance between the hand trajectory and the straight line joining the start and end targets.

**Force compensation.**    On every channel trial (error-clamp trial), the force compensation was calculated to quantify the amount of predictive force adaptation to the force field. Force compensation is defined on each trial by the regression between the force produced by participants into the wall of the simulated channel (lateral measured force) and the force needed to compensate perfectly for the force field [16]. The perfect compensatory force can be determined as the forward velocity of each trial multiplied by the force field constant. As experiments were counterbalanced across participants, the values were flipped such that the force compensation associated with each cue has the same sign for every participant. All models used to fit the results of the data assume equal adaptation to both force fields. In order to allow the models to fit the data, we subtracted blockwise the mean force compensation of both contextual cues from the force compensation values for each contextual cue (Figs 3B and 6B). Therefore the mean compensation across both contextual cues for the whole experiment is zero. Additionally, figures of the natural force compensation (without the mean force compensation subtracted) were added in the Supporting Information (S1 Fig).

**Force profiles.**   In order to examine the shape and timing of the predictive forces on channel trials during the experiment, the force profile in the x-axis was normalized by the perfect compensation force profile. This perfect force profile was calculated as the y-axis velocity multiplied by the force field constant.

Throughout the experiments, data are primarily presented for each block. However, a different number of trials (blocks) were performed by each participant in the de-adaptation phase. For plotting purposes, this phase was divided into equal-sized sections, where the mean data was determined for each section rather than for each block in order to allow averaging across participants. For kinematic error twenty sections were used, whereas for force compensation seven sections were used.

**Decay.**   Our experimental design was set to explore memory decay both throughout the error-clamp phase and within the adaptation phase by having 10 decay blocks during this phase. In experiment 1, within a decay block, four additional channel trials were included directly after the usual channel trial for one of the cues, such that a row of 5 channel trials in a row were created (3 channel trials in experiment 2). A single channel trial for the other cue was introduced normally within these blocks. The specific cue for each of the channel blocks alternated, with a total of five blocks performed for each of the cues. In order to compare the decay within the decay blocks and error-clamp phases, we simulated the effect of range of decay rates on a previously learned process. The decay was simulated such that

$$x^{n+1} = A \cdot x^n \tag{2}$$

with the retention rate A being set as 0.5, 0.8, 0.9, 0.95, 0.98, 0.993, 0.997, 0.999 or 1. We chose a starting value corresponding to the level of learning of experiment 2 for comparison purposes. The comparison of experimental data with different simulated decay rates allows them to be associated with the respective rates of a slow, "ultraslow" or "hyperslow" process, and for comparison independent of any specific model.

**Statistics.**   Using the statistical software JASP (version 0.9.2), repeated measures ANOVAs were run on the kinematic error and the force compensation, with factors *Stage* (4 levels) and *Cue* (2 levels). For comparisons of kinematic error we compared across 4 stages (first 5 blocks in adaptation; final 10 blocks in adaptation; first 5 blocks in de-adaptation; final 5 blocks in de-adaptation). For comparisons of force compensation we compared across a different 4 stages (final 5 blocks in pre-exposure; final 10 blocks in adaptation; final 3 blocks of de-adaptation; all blocks of error-clamp phase). For both ANOVAs the sign of the values for one of the cues was flipped in order to examine whether there were differences in the levels of adaptation between cues. If the main effects were significant ($p < 0.05$), post hoc comparisons were performed using the Bonferroni test. In order to examine specific differences in the level of force compensation between the cues, paired t-tests were performed.

## Simulations

In order to investigate the computational mechanisms underlying the evolution of dual-adaptation and the formation of motor memories we modeled motor adaptation to two opposing force fields each associated to a contextual cue. For our simulations we examine two models out of a family of learning-from-error equations based on the dual-rate model [16]:

$$x^{n+1} = A \cdot x^n + B \cdot e^n$$
$$x^n = f^n - x^n \tag{3}$$

where:

$x^{n+1}$ – output on subsequent trial

$x^{n+}$ – output on current trial

$e^n$ – error on current trial

$A$ – retention rate

$B$ – learning rate

$f^n$ – environmental force

and adaptation occurs through the summation (and competition) of two separate states with different timescales of learning and retention.

In order to model dual-adaptation, we simulate both the "one-fast-two-slow-binary-switch" model [24], which we use as our baseline model for comparison:

$$
\begin{aligned}
x^{n+1} &= x_f^{n+1} + \boldsymbol{x}_s^{n+1} \cdot \boldsymbol{c}^{n+1} \\
x_f^{n+1} &= A_f \cdot x_f^n + B_f \cdot e^n \\
\boldsymbol{x}_s^{n+1} &= A_s \cdot \boldsymbol{x}_s^n + B_s \cdot e^n \cdot \boldsymbol{c}^n
\end{aligned}
\tag{4}
$$

and a "two-fast-two-slow-binary-switch" model:

$$
\begin{aligned}
x^{n+1} &= (\boldsymbol{x}_f^{n+1} + \boldsymbol{x}_s^{n+1}) \cdot \boldsymbol{c}^{n+1} \\
\boldsymbol{x}_f^{n+1} &= A_f \cdot \boldsymbol{x}_f^n + B_f \cdot e^n \cdot \boldsymbol{c}^n \\
\boldsymbol{x}_s^{n+1} &= A_s \cdot \boldsymbol{x}_s^n + B_s \cdot e^n \cdot \boldsymbol{c}^n
\end{aligned}
\tag{5}
$$

on an A-B-Error-clamp paradigm similar to our experiment. For our simulations, the parameters were taken from Smith and colleagues (2006) [16], with $A_f = 0.92$ and $B_f = 0.05$ for the fast process and with $A_s = 0.999$ and $B_s = 0.01$ for the slow process, and from Lee & Schweighofer (2009) [24], $\boldsymbol{c}$ being the contextual cue switch parameter. Because the authors assumed no interference and perfect switching among internal states in a process, we used a unit vector $\boldsymbol{c}^n = [0\ 1]$.

## Model fitting

The experimental data was fit by twelve models from a family of learning-from-error equations (Eq 1) for each participant individually. To assess the models, the mean of the individual fits were compared. A previous study [24] assumed the contextual switch (c) between the cues was binary. However, other models of sensorimotor learning such as the MOSAIC model [2,74], have suggested a responsibility estimator that would provide a weighted mixture between modules. In order to examine both of these possibilities, we set the contextual switch as binary for half of the models (c set as [0 1]) and as a parameter for the other half of the models (c set as [c 1-c]).

Specifically, we fit twelve models with parallel organization between the fast and slow processes. These consisted of a dual-rate, a triple-rate and a quadruple-rate model where each could be built with either one fast process or multiple fast processes and where the contextual cue switch was either binary or weighted. The general equation of these models can be written:

$$
x^{n+1} = x_f^{n+1} + (\boldsymbol{x}_s^{n+1} + \boldsymbol{x}_{us}^{n+1} + \boldsymbol{x}_{hs}^{n+1}) \cdot \boldsymbol{c}^{n+1}
\tag{6}
$$

for models containing a single fast process or

$$
x^{n+1} = (\boldsymbol{x}_f^{n+1} + \boldsymbol{x}_s^{n+1} + \boldsymbol{x}_{us}^{n+1} + \boldsymbol{x}_{hs}^{n+1}) \cdot \boldsymbol{c}^{n+1}
\tag{7}
$$

for models containing multiple fast processes.

Therefore, the fast process contains either a single state and is written as

$$x_f^{n+1} = A_f \cdot x_f^n + B_f \cdot e^n \tag{8}$$

or multiple states and is written as

$$\boldsymbol{x}_f^{n+1} = A_f \cdot \boldsymbol{x}_f^n + B_f \cdot e^n \cdot \boldsymbol{c}^n \tag{9}$$

The slow, "ultraslow" and "hyperslow" processes [37] contain multiple states and are written:

$$\boldsymbol{x}_s^{n+1} = A_s \cdot \boldsymbol{x}_s^n + B_s \cdot e^n \cdot \boldsymbol{c}^n \tag{10}$$

$$\boldsymbol{x}_{us}^{n+1} = A_{us} \cdot \boldsymbol{x}_{us}^n + B_{us} \cdot e^n \cdot \boldsymbol{c}^n \tag{11}$$

$$\boldsymbol{x}_{hs}^{n+1} = A_{hs} \cdot \boldsymbol{x}_{hs}^n + B_{hs} \cdot e^n \cdot \boldsymbol{c}^n \tag{12}$$

with

$x_{hs}^n = 0$ for the dual-rate models,

$x_{hs}^n = 0$ for the dual-rate and triple-rate models.

Here the hyperslow process has a lower learning rate and higher retention rate than the ultraslow process. The one-fast-two-slow-binary-switch model [24] was used as our referent model (Eq 4).

Each model was fit to the force compensation data throughout an entire experiment. In order to fit the models, the mean force compensation values over the two cues was subtracted to remove any potential bias between the cues (we also performed the same analysis without this subtraction; S3 Fig, S4 Fig, S5 Fig in the Supporting Information). For each model, we found the parameter values that best fit the experimental data, using a least-squares method (fminsearchbnd), where the parameters were constrained:

$$0.5 < A_f < 0.95 < A_s, A_{us} < A_{hs} < 1$$

$$0.1 < B_f < 0.35$$

$$0 < B_{hs}, B_{us} < 0.02 < B_s < 0.35$$

$$B_s < B_f$$

$$0.5 < c < 1$$

The optimization was performed ten times for each fit, with a random initial parameter setting within the parameter constraints, and the one with the smallest error used. The values for these constraints were determined by examining the results of previous studies (Table 2). The learning rates were constrained within specific bounds in order to ensure that each temporal process corresponds to the appropriate scale as determined from prior literature. For example, a slow process must be constrained within a valid range of possible learning rates, otherwise it could be used to fit a hyperslow or fast process within the optimization procedure. The retention values for the slow, ultraslow and hyperslow processes were only constrained to be larger than 0.95, a value far below all the results of previous studies (all above 0.99, Table 2). However both parameters for the fast process were constrained. As the purpose of this experiment was to determine whether there exists a single or multiple fast processes, we required that the

parameters of this fast process are within the range of previous studies (Table 2). If this was not constrained, then the optimization routine could potentially fit the slow process parameters to the fast process, rather than find a single fast process for specific models. This allows a fair comparison of the fits across all models. After consideration of the results of previous studies (Table 2) we used parameter constraints that incorporate all of the previously found parameters.

In order to compare across the models, Bayesian information criterion (BIC) was calculated and used to select the preferred model (lowest BIC). If the difference between two BIC values is between 0 to 2 there is considered to be no difference between the models, between 2 to 6 the difference is considered to be positive (small but existing), between 6 to 10 the difference is considered to be strong, and a difference exceeding 10 is described as very strong [75]. Specifically we calculated the improvement in BIC value, defined as the difference between the BIC value of the referent model and the other models. A BIC improvement of greater than 6 therefore indicates a strong improvement compared to the one-fast-two-slow-binary-switch model. Both the individual BIC improvements for each participant, and the mean BIC improvements across participants are reported.

**Model recovery analysis.** To assess the reliability of the model fitting, we included a model-recovery analysis [76] using simulated data based on our experimental design (both experiment 1 and 2). For each participant's experimental design, we simulated eight noisy sets of data for each of the twelve models, resulting in 80 datasets per model per experiment. The level of noise was estimated from the mean variability in force compensation of the 10 participants during the second half of the pre-exposure phase. The simulated noise was uniformly distributed between ± 0.0831 in experiment 1 and ± 0.0715 in experiment 2. We then fit all twelve models to each simulated dataset (identical to the fitting of experimental data) and estimated which model is most likely to have generated the data using BIC. Finally, we compare the best-fit models to the real models that were used to simulate the data in a confusion matrix showing the probability of a model to fit the generated data (S7 Fig) and the average BIC differences relative to the best-fit models (S8 Fig).

The results of the model-recovery analysis (S7 Fig) indicate that the model fitting is able to well capture the type of model in terms of the number of fast-processes or whether the contextual cue switch is weighted or binary, but often selects the models with fewer timescales (e.g. triple-rate rather than quadruple-rate).

## Supporting information

**S1 Fig. Force compensation without subtraction of mean force.** Mean of force compensation over pre-exposure (white), adaptation (grey), de-adaptation (dark grey) and error-clamp (light grey) phases. The mean force compensation across the two cues is not subtracted from the force compensation. The data of the contextual cue 1 (left visual workspace shift) and 2 (right visual workspace shift) are presented in red and blue lines, respectively. Shaded regions indicate the standard-error of the mean. **A.** Experiment 1. **B.** Experiment 2. (TIF)

**S2 Fig. Comparison between one-fast and two-fast models.** Bayesian Information criterion (BIC) differences between the one-fast-two-slow-dual-rate model (binary switch, left side) and the two-fast-two-slow-dual rate model (binary switch, right side). Improvements in BIC from 2 to 6, 6 to 10 and greater than 10 are considered as a positive, strong and very strong evidence of a model fitting better than the other models, respectively. The red dashed lines show a BIC difference of 6, indicating strong evidence towards one of the models. **A.** Experiment 1. **B.** Experiment 2. (TIF)

**S3 Fig. BIC model comparison and frequency table for experiment 1 without subtraction of mean force.** The fitting is run over the force compensation without subtraction of the mean between cues. **A.** Frequency table for each model (y-axis) by priority order (x-axis). This table represents the number of participants in which a given model was selected as the best-fit to the participant's data by BIC from first chosen, to twelfth (last) chosen. On the right-side, an opacity scale represents the number of participants. **B.** Individual BIC improvement for model comparison. The twelve models are differentiated by their improvement in BIC relative to the referent model (value of 0, left). For each participant, models are normalized to reveal the qualitative differences between each model and the best-fit model for that participant (value of 1). On the right side, the BIC improvement relative to the best-fit model is shown for the unnormalized values. Improvements in BIC from 2 to 6, 6 to 10 and greater than 10 are considered as a positive, strong and very strong evidence of a model fitting better than the other models, respectively. The red dashed line shows a BIC difference of 6 from the best-fitting model. (TIF)

**S4 Fig. BIC model comparison and frequency table for experiment 2 without subtraction of mean force.** The fitting is run over the force compensation without subtraction of the mean between cues. Plotted as experiment 1. **A.** Frequency table for each model (y-axis) by priority order (x-axis). The right-side opacity scale represents the number of participants. **B.** Individual BIC improvement for model comparison. (TIF)

**S5 Fig. BIC frequency table across experiments.** Frequency table of the individuals Bayesian Information criterion (BIC) model according to their priority (order of preference according to BIC improvement) across both experiments 1 and 2. The fitting is run over the force compensation without subtraction of the mean between cues. (TIF)

**S6 Fig. Simulation of best-fit set of parameters (median across participants) for the twelve models.** Here, the de-adaptation phase contained 220 trials (intermediate between experiment 1 and 2). The temporal phases are defined as pre-exposure (white), adaptation (grey), de-adaptation (dark grey) and error-clamp (light grey) phases. The data of the contextual cue 1 (left visual workspace shift) and 2 (right visual workspace shift) are presented in red and blue lines, respectively. The total output for each contextual cue (dark red and dark blue lines) is composed of the summation of each process (light red and light blue lines): hyperslow, ultraslow and slow processes (long, medium and short dashed lines respectively) and fast processes (solid lines). When a single fast process is shared between cues, this is represented with a magenta line. Note that spontaneous recovery is revealed for both contextual cues in the error-clamp phase for triple and quadruple-rate models. (TIF)

**S7 Fig. Confusion matrices from the model recovery analysis. A.** Experiment 1. **B.** Experiment 2. These matrices show the effect of prior parameter distributions on model recovery. Numbers denote the probability out of 100 repetitions (color scale) that data generated with model X are best fit by model Y, thus the confusion matrix represents p(bestfit model | real model). (TIF)

**S8 Fig. BIC difference confusion matrices from the model recovery analysis. A.** Experiment 1. **B.** Experiment 2. The values represent the average BIC differences relative to the best model. These matrices show the effect of prior parameter distributions on model recovery. Numbers

denote the mean difference in BIC between this model and the best selected model across the 100 repetitions. If a specific model was selected as the best-fit model all 100 times, then this mean BIC difference would be zero. In all other cases, the values would be negative. This negative value corresponds to the average difference in BIC between each repetition and the best fit (value of 0). This table adds additional information on model ordering, as the confusion matrices (S7 Fig) do not take into account the magnitude of the BIC difference. We can see that the diagonal is always close to the best-fit model in terms of BIC even when it is not most often selected.
(TIF)

## Acknowledgments

We thank Justinas Česonis for helpful suggestions throughout this work.

## Author Contributions

**Conceptualization:** Marion Forano, David W. Franklin.

**Formal analysis:** Marion Forano, David W. Franklin.

**Investigation:** Marion Forano.

**Methodology:** Marion Forano, David W. Franklin.

**Writing – original draft:** Marion Forano.

**Writing – review & editing:** Marion Forano, David W. Franklin.

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
