## [Decision Letter · Decision Letter 0]

7 Mar 2020

Dear Mrs. Forano,

Thank you very much for submitting your manuscript "Timescales of motor memory formation in dual-adaptation" for consideration at PLOS Computational Biology.

As with all papers reviewed by the journal, your manuscript was reviewed by members of the editorial board and by several independent reviewers. The reviewers were in general positive about the paper, but highlighted a number of concerns, mostly relating to the computational modeling and interpretation. In particular, the reviewers highlighted that the model fits are not presented in a transparent manner (only plotting BIC scores), despite it being seemingly straightforward to visualize predictions of this type of model. In addition, in order to improve the rigor of the modeling component of the paper, it would be worthwhile to include a model-recovery analysis using simulated data to determine how reliably the different models you consider can be discriminated based on the type of data collected in the experiments (see e.g. Wilson and Collins, eLife, 2019). In light of the reviews (below this email), we would like to invite the resubmission of a significantly-revised version that takes into account all the reviewers' comments.

We cannot make any decision about publication until we have seen the revised manuscript and your response to the reviewers' comments. Your revised manuscript is also likely to be sent to reviewers for further evaluation.

Sincerely,

Adrian M Haith

Associate Editor

PLOS Computational Biology

Samuel Gershman

Deputy Editor

PLOS Computational Biology

Reviewer's Responses to Questions

**Comments to the Authors:**

Reviewer #1: My comments are attached separately.

Reviewer #2: This manuscript from Forano and Franklin tackles the question of timescales of motor memory formation during dual motor adaptation. Previous groups have shown that interference can be minimized and separate memories can be formed during simultaneous adaptation to opposing perturbations when participants are provided with appropriate contextual cues (Lee and Schweighofer 2009, Howard et al 2012, 2013, and 2015). This phenomenon was modeled by Lee and Schweighofer through expansion of a two-state multi-rate model of adaptation (Smith et al 2006), such that separate slow-learning processes were each assigned to one of the two perturbations, but only a single fast process adapted to opposing perturbations. Using simulation, two behavioral experiments, and model selection, the authors of the present manuscript propose a different computational architecture to that of Lee and Schweighofer. Specifically, they propose separate fast and slow processes for each perturbation, as well as a perturbation-specific “ultra-slow” process.

Overall, I thought many parts of this manuscript were clearly written and the analyses were rigorous. However, I have major concerns regarding the narrow scope of the research question and thus the authors’ attempt to frame their findings within the broader context of skill learning, as opposed to the more constrained realm of adaptation (re-calibration). At the end of the Introduction, the primary aim of the study was articulated as a question about whether or not the Lee and Schweighofer model could explain spontaneous recovery (the return to an adapted state during error clamp trials following washout of adaptation) during dual adaptation. While this question of interest, the link to skill learning (e.g., learning to ride a bicycle or ski) appears tenuous, especially in light of research showing that skill learning and recalibration are qualitatively different learning phenomena (Telgen et al 2014, Yang et al 2020 bioRxiv).

Other related Major Concerns follow:

- The addition of the error clamp trials to the AB experimental design was well-justified and had a clear link to the primary research question. However, I did not understand the justification for the subtle differences in experimental design between E1 and E2 and the main text did not provide a clear rationale. This hampered the overall impact of the manuscript as two similar sets of results and analyses were reported sequentially, but without any clear context for why both versions of the experiment were performed. For clarity, could the authors either provide a better justification for the two experimental designs and/or provide the rationale for combining the experimental results earlier so that readers only need to deal with one set of results? I believe this would help with clarity, as there were not independent pieces of information gained from both experiments.

-I was surprised by the lack of explicit discussion of how the current results relate to the series of papers from Howard and Wolpert on forming separate internal models. I’m curious as to what the authors think regarding the generalizability of their findings and models to other methods of blocking interference, specifically dynamic contextual cues. This is in my mind of interest because the methods used in the current manuscript seem radically different than theirs, yet the phenomenon of interest is the same.

In addition, if we are to accept the findings of the Howard et al and Sheahan et al papers, don’t their behavioral results already speak to the formation of two separate internal models? My point being that if we assume that each internal model has a slow and fast process, then the current proposition of two fast processes for two separate contexts is a natural corollary of what has already been shown in the literature. This is not to take away from the current findings and the more formalized approach, and of course, the introduction of a third ultra-slow process, but it does seem worth discussion in the text. This concern also points to the narrow scope of the manuscript.

-Since the bulk of the interpretation rests on the model-fitting procedures, could the authors provide figures showing model fits, at least for the models with characteristically different architectures, e.g., one-fast vs two-fast and two-slow vs two-fast and three-slow? I think such a figure would help contextualize the magnitude of the differences between the models, as I suspect the differences between two-fast two-slow vs two-fast three-slow would be fairly subtle as compared to one-fast vs two-fast. But without such a figure, the reader is only left to work with the BIC scores, which are informative, but not nearly as impactful as a figure with model fits could be.

-While the figures showing the individual and group-averaged BIC results were both attractive and informative, I am concerned that too much emphasis is placed on group averaged scores, as there was no measure of across individual variance shown in Figs. 4 or 6. The mean improvements without the standard deviation or error tells only part of the story. Also, wouldn’t it be justified to treat the BIC scores as any other dependent variable and subject them to inferential statistics, too? My point is that the BIC scores may benefit a great deal from contextualizing with figures of model fits and statistical tests. Since you are not testing different classes of models or radically different architectures, but rather different versions of multi-rate models that share a common algorithm, these suggestions would help the reader form a more informed opinion regarding the robustness of the results.

On a related note, why does it appear that there are no normalized BIC scores for the orange and red models for participants 8 and 9 in Fig. 4B?

-In terms of the writing, the Introduction and Results would benefit from more concrete details regarding some of the key concepts being introduced. For example, in the third paragraph, there is much discussion of contextual cues, but no delineation of specific cues that have been used in the literature and why some may be more effective than others (e.g., lead-ins or follow-throughs versus static color cues). Even as a member of the motor learning community, I was left wondering what you were specifically referring to. The overall effect is that the writing comes off as dry and likely impenetrable to someone outside of our small discipline. Also, as mentioned earlier, there does not appear to be a justification for E2, which also presents a barrier to the reader fully engaging with the study as currently presented.

-If you are proposing an ultra-slow process, shouldn’t the parameter for A_us be constrained such that A_us > A_slow? I see in Table 1 that A_us is actually lower than A_slow. This doesn’t seem to follow the logic of having separate processes acting on different timescales, i.e., learning slower and retaining more or vice versa. I’m not sure how much this would affect the results, but some explanation seems appropriate.

-Why was the mean force compensation across the two cues subtracted from the raw force compensation? And why process only forces this way, as it appears kinematic errors were left alone?

-Although the “stages” and “cues” used in the ANOVAs were clearly explained, there was no mention of what specific movement cycles were used in the analyses in the main text. This seems important to report as the effect sizes for spontaneous recovery do not appear very large.

-The paragraph beginning on line 499 is a clear example of conflating adaptation with skill learning. Perhaps either flesh out the connection you would like to make in order to make the analogy to real world activities more convincing, or consider supplying appropriate caveats to your current interpretation.

Minor concerns:

-The A and B panels are flipped in Figure 2.

-Caption for Figure 1 mentions contextual cues 1 and 2, but there has still been no mention of what these cues are.

-There’s a word or two missing in lines 545-548. This sentence overall is difficult to understand.

Reviewer #3: This study investigated the memory structure when adapting to the opposing dynamics during reaching movements. A previous study (Lee & Schweighofer, J Neurosci 2009) has proposed that the motor memory for the opposing visual rotation has a single fast and double slow motor memory. The authors tested the validity of this model by employing A-B-Error clamp experimental paradigm within dual-task adaptation. Contrary to the prediction by the previously proposed model, the authors observed the spontaneous recovery, clearly indicating the presence of two-fast memory states. They also fit various types of multi-rate models to the data and found the model with two fast, slow, and ultra-slow time constants best explained the data. This is an interesting study demonstrating a new aspect of motor memory structure, and the experiment and analysis were well designed. As described below, however, I would like to raise several issues, especially for the interpretation of the motor memory structure.

Major points

1) I was confused about the idea that motor memory has a discrete number of processes (e.g., two fast and slow processes). What does it mean? In the current experiment, the distance between the starting positions (i.e., visual workspace) was 20 cm. However, if the distance was 0 cm, different motor memories cannot be created. These facts indicate that different motor memories are likely to be created depending on the visual workspace. If it is the case, an infinite number of discrete processes seems necessary. The presence of two processes shown in this study is an epiphenomenon caused by the experimental design using just two contextual cues? How reasonable is assuming the memory has just two processes?

2) The more natural idea would be to assume that the motor memories are composed of fast and slow (and ultra-slow) motor primitives (Thoroughman & Shadmehr, Nature 2000), and these motor primitives are recruited depending on visual workspaces. Then, the problem to be questioned here should be how the primitives are recruited rather than the number of processes. I think that the model proposed in the present study is the approximation of such a motor primitive based model (i.e., the parameter ‘c’ is likely to reflect the way of encoding of visual workspace in the primitives). It seems a little bit odd to me that the motor memory has just two processes, and the contribution of each process is modulated with the visual cue. I do not think the authors need to change the whole story, but I would suggest that the authors mention the discussion on this issue.

Minor points

1) Line 216: The comma should be removed?

2) Line 551: The “)” seems necessary after “[38]”.

3) Line 787: “In” should be removed?

**Have all data underlying the figures and results presented in the manuscript been provided?**

Reviewer #1: None

Reviewer #2: Yes

Reviewer #3: Yes

PLOS authors have the option to publish the peer review history of their article (what does this mean?). If published, this will include your full peer review and any attached files.

Reviewer #1: No

Reviewer #2: No

Reviewer #3: No
---

## [Decision Letter · Decision Letter 1]

19 Jul 2020

Dear Mrs. Forano,

Thank you very much for submitting your manuscript "Timescales of motor memory formation in dual-adaptation" for consideration at PLOS Computational Biology.

As with all papers reviewed by the journal, your manuscript was reviewed by members of the editorial board and by several independent reviewers. The revisions have substantially improved the paper, and the reviewers were mostly satisfied that their concerns have been resolved. However, Reviewer 1 raised some valid concerns regarding parameter bounds used in fitting the model and whether these might influence the conclusions of the paper. In light of the reviews (below this email), we would like to invite the resubmission of a significantly-revised version that takes into account the reviewers' comments.

We cannot make any decision about publication until we have seen the revised manuscript and your response to the reviewers' comments. Your revised manuscript is also likely to be sent to reviewers for further evaluation.

Sincerely,

Adrian M Haith

Associate Editor

PLOS Computational Biology

Samuel Gershman

Deputy Editor

PLOS Computational Biology

Reviewer's Responses to Questions

**Comments to the Authors:**

Reviewer #1: review uploaded as attachment

Reviewer #2: I found the revised manuscript to be much clearer and easier to read than the original submission, and I appreciate the changes the authors made based on our suggestions. The additional discussion of previous literature related to contextual interference and ideas surrounding multiple paired models helped me better contextualize the current findings. In addition, the parameter recovery analysis and new figures strengthened the authors’ overall interpretation of their findings.

Below are additional suggestions and minor comments for the authors to consider:

- I suggest redoing Fig. S5 and including a revised version in the main text. Due to the compression of the x-axes, the simulations are not nearly as clear or informative as they could be. Have the authors considered selecting a subset of their models and showing them in a more expanded form in the main text? The remaining models (or all of them) could remain a supplementary figure, with fewer columns and more rows.

Minor:

- The font in the tables is too small and difficult to read without zooming in.

- The language used to describe best-fit models was not always clear. For example, lines 285-286: “…out of ten participant, nine participants equally chose the…” The italicized portion is what I’m referring to. Also, here and throughout, the authors describe participants “choosing” their models. Wouldn’t it be more accurate, and less idiosyncratic, to write in these cases something like “…the X model provided the best fit to a out of b participants.”

- L381: I would replace “deeper” with “greater” or “more”. I wasn’t clear about what “deeper” meant until looking at the figure.

- L551: This sentence was difficult to parse, especially the reference to “in the concept of single adaptation”.

Reviewer #3: I am satisfied with the authors’ explanation regarding the interpretation of multiple motor memories. I have no further concerns.

**Have all data underlying the figures and results presented in the manuscript been provided?**

Reviewer #1: Yes

Reviewer #2: None

Reviewer #3: None

PLOS authors have the option to publish the peer review history of their article (what does this mean?). If published, this will include your full peer review and any attached files.

Reviewer #1: No

Reviewer #2: No

Reviewer #3: No
---

## [Decision Letter · Decision Letter 2]

9 Sep 2020

Dear Mrs. Forano,

We are pleased to inform you that your manuscript 'Timescales of motor memory formation in dual-adaptation' has been provisionally accepted for publication in PLOS Computational Biology.

Note that Reviewer 1 had one remaining request for clarification which you may wish to address in your final version.

Best regards,

Adrian M Haith

Associate Editor

PLOS Computational Biology

Samuel Gershman

Deputy Editor

PLOS Computational Biology

Reviewer's Responses to Questions

**Comments to the Authors:**

Reviewer #1: My comments are uploaded as an attachment.

Reviewer #2: I am satisfied with all of the changes made to the manuscript and have no further suggestions.

**Have all data underlying the figures and results presented in the manuscript been provided?**

Reviewer #1: None

Reviewer #2: Yes

PLOS authors have the option to publish the peer review history of their article (what does this mean?). If published, this will include your full peer review and any attached files.

Reviewer #1: No

Reviewer #2: No

---

## [Editor Report · Acceptance letter]

13 Oct 2020

PCOMPBIOL-D-20-00086R2 

Timescales of motor memory formation in dual-adaptation

Dear Dr Forano,

I am pleased to inform you that your manuscript has been formally accepted for publication in PLOS Computational Biology. Your manuscript is now with our production department and you will be notified of the publication date in due course.

With kind regards,

Laura Mallard
